# *Atrx* inactivation drives disease-defining phenotypes in glioma cells of origin through global epigenomic remodeling

Carla Danussi[1], Promita Bose[2], Prasanna T. Parthasarathy[2], Pedro C. Silberman[2], John S. Van Arnam[3], Mark Vitucci[4], Oliver Y. Tang [2], Adriana Heguy[5], Yuxiang Wang[2], Timothy A. Chan [2,6], Gregory J. Riggins[7], Erik P. Sulman [1,8], Frederick F. Lang[9], Chad J. Creighton[10], Benjamin Deneen [11], C. Ryan Miller[4,12], David J. Picketts[13,14], Kasthuri Kannan [5] & Jason T. Huse[1,3]

Mutational inactivation of the SWI/SNF chromatin regulator *ATRX* occurs frequently in gliomas, the most common primary brain tumors. Whether and how ATRX deficiency promotes oncogenesis by epigenomic dysregulation remains unclear, despite its recent implication in both genomic instability and telomere dysfunction. Here we report that *Atrx* loss recapitulates characteristic disease phenotypes and molecular features in putative glioma cells of origin, inducing cellular motility although also shifting differentiation state and potential toward an astrocytic rather than neuronal histiogenic profile. Moreover, Atrx deficiency drives widespread shifts in chromatin accessibility, histone composition, and transcription in a distribution almost entirely restricted to genomic sites normally bound by the protein. Finally, direct gene targets of Atrx that mediate specific Atrx-deficient phenotypes in vitro exhibit similarly selective misexpression in *ATRX*-mutant human gliomas. These findings demonstrate that ATRX deficiency and its epigenomic sequelae are sufficient to induce disease-defining oncogenic phenotypes in appropriate cellular and molecular contexts.

[1] Department of Translational Molecular Pathology, University of Texas MD Anderson Cancer Center, Houston, TX 77030, USA. [2] Human Oncology and Pathogenesis Program, Memorial Sloan-Kettering Cancer Center, New York, NY 10065, USA. [3] Department of Pathology, University of Texas MD Anderson Cancer Center, Houston, TX 77030, USA. [4] Department of Pathology and Laboratory Medicine, University of North Carolina School of Medicine, Chapel Hill, NC 27516, USA. [5] Department of Pathology, New York University School of Medicine, New York, NY 10016, USA. [6] Department of Radiation Oncology, Memorial Sloan-Kettering Cancer Center, New York, NY 10065, USA. [7] Departments of Neurosurgery, Oncology, and Genetic Medicine, Johns Hopkins School of Medicine, Baltimore, MD 21231, USA. [8] Department of Radiation Oncology, University of Texas MD Anderson Cancer Center, Houston, TX 77030, USA. [9] Department of Neurosurgery, University of Texas MD Anderson Cancer Center, Houston, TX 77030, USA. [10] Department of Medicine and Dan L. Duncan Comprehensive Cancer Center Division of Biostatistics, Baylor College of Medicine, Houston, TX 77030, USA. [11] Department of Neuroscience, Baylor College of Medicine, Houston, TX 77030, USA. [12] Departments of Pharmacology and Neurology, Lineberger Comprehensive Cancer Center and Neuroscience Center, University of North Carolina School of Medicine, Chapel Hill, NC 27516, USA. [13] Department of Biochemistry, Microbiology, and Immunology, University of Ottawa, Ottawa, ON K1H 8L6, Canada. [14] Ottawa Hospital Research Institute, Ottawa, ON K1H 8L6, Canada. Correspondence and requests for materials should be addressed to J.T.H. (email: jhuse@mdanderson.org)

Somatic mutations in genes encoding epigenetic regulators have now been widely reported across a number of cancer types[1]. Diffusely infiltrating gliomas, the most common primary brain tumors, feature loss-of function mutations in the SWI/SNF chromatin remodeler gene *ATRX* (α-thalassemia mental retardation X-linked) as defining molecular alterations delineating major adult and pediatric disease subtypes[2–5]. These incurable glioma variants tend to exhibit morphological and immunohistochemical features of astrocytes and, accordingly, are classified as "astrocytomas". Within this disease context, ATRX deficiency invariably co-occurs with mutations in *TP53* and in genes encoding either isocitrate dehydrogenase enzymes (*IDH1*

and *IDH2*) in adults or H3.3 histone monomers (*H3F3A* and *HIST1H3B*) in children. However, inactivating *ATRX* mutations have also been identified in neuroblastoma, pancreatic neuroendocrine tumor, and multiple sarcomas, where they arise in distinct molecular and cellular contexts, indicating that oncogenic mechanisms of broad physiological relevance are likely mobilized by ATRX deficiency[6].

What these mechanisms precisely entail remains unclear. Germline mutations in *ATRX* cause a rare, congenital neurodevelopmental condition associated with intellectual disability (ATR-X syndrome)[7], which likely results from extensive p53-dependent apoptosis within the neuronal precursor

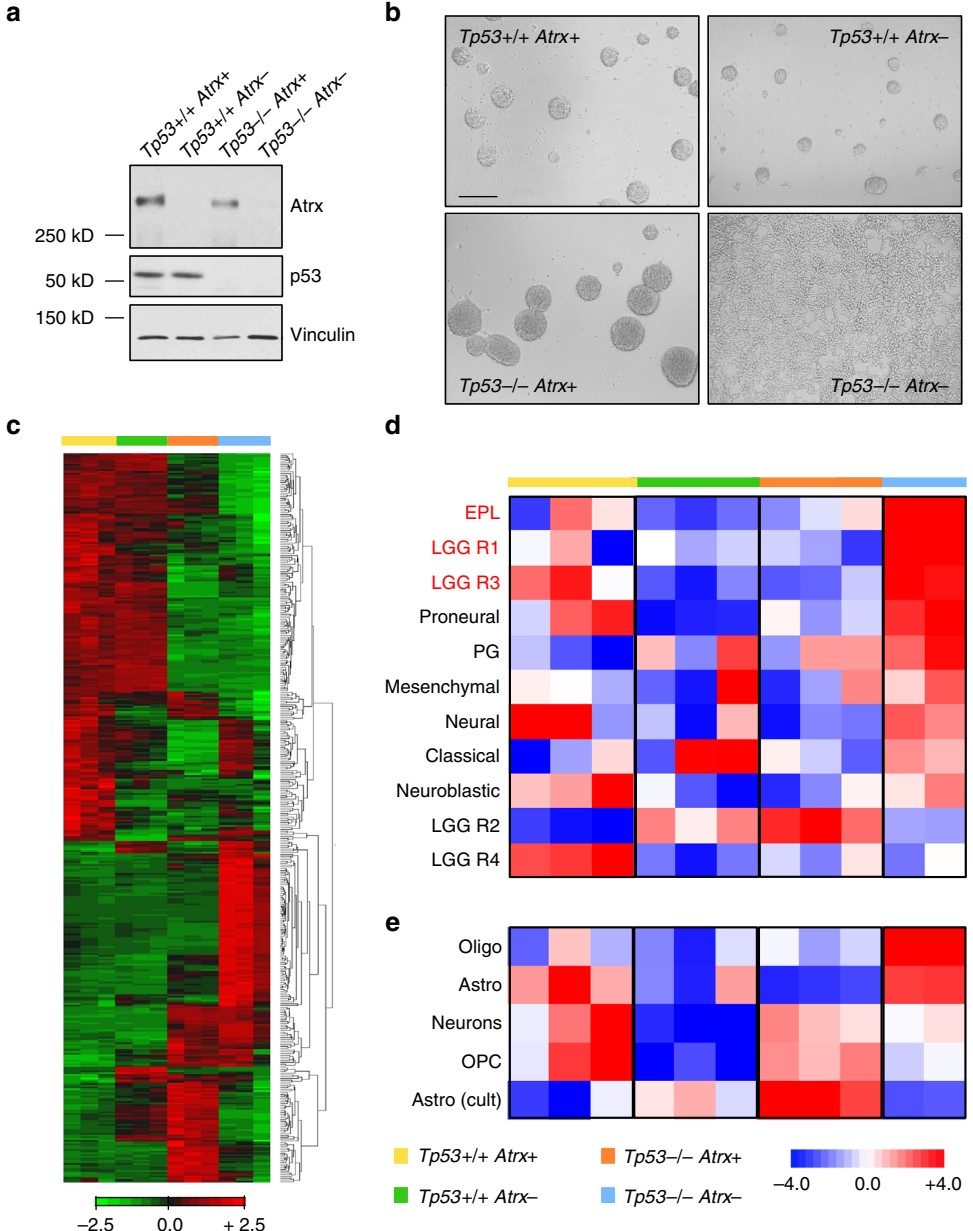

**Fig. 1** Atrx deficiency alters the morphology and gene expression profiles of mNPCs. **a** Western blot demonstrating effective cre-recombinase-mediated Atrx inactivation. p53 levels are also shown (Vinculin loading control). **b** micrographs showing mNPC morphology at 5 passages following Atrx inactivation (representative of three biological replicates; scale bar 100 μm). **c** heat map showing differentially expressed genes across isogenic mNPCs (Tp53$^{+/+}$ Atrx+ : yellow, Tp53$^{+/+}$ Atrx−: green, Tp53$^{-/-}$ Atrx+ : orange, and Tp53$^{-/-}$ Atrx−: blue). **d**, **e** heat map showing ssGSEA correlations between mNPC expression profiles and transcriptional signatures derived from glioma subtypes (**d**) and mature murine nervous system constituents (**e**). Glioma subtypes exhibiting high rates of ATRX deficiency shown in red. EPL early progenitor-like, PG pre-glioblastoma, Oligo oligodendrocyte, Astro astrocyte, OPC oligodendroglial precursor cell, Astro (cult) cultured astrocyte

compartment[8]. Such cell death may derive from genomic instability, given that ATRX deficiency has been repeatedly linked to replication stress, DNA damage, and aneuploidy[6]. Recent work suggests that the deleterious effects of ATRX deficiency on genomic integrity induce the alternative lengthening of telomeres (ALT) phenotype exhibited by tumors harboring *ATRX* mutations[9,10]. ALT is a telomerase-independent mechanism of telomere maintenance based on homologous recombination[11], and as such, may serve to potentiate cellular immortality. However, while the potential for unlimited self-renewal obviously represents a requirement for oncogenesis, its sufficiency in and of itself is questionable. By this reasoning, ATRX deficiency must induce additional oncogenic phenotypes in affected tumors, likely involving shifts in transcriptional profiles, given the established role of ATRX in chromatin biology.

Recent studies have extensively implicated ATRX in the regulation of chromatin state and composition[12–15]. In particular, ATRX is known to form a complex with DAXX (death-associated protein 6) to modulate H3.3 histone composition at sites across the genome[16,17]. ATRX also appears to be required for the normal expression of specific genes. For instance, impaired ATRX-dependent transcription at the α-globin locus is thought to induce α-thalassemia in ATR-X syndrome[18]. Whether and how the epigenomic consequences of ATRX deficiency promote gliomagenesis and disease-relevant phenotypes has not been explored.

To model the cellular and molecular context of *ATRX*-mutant gliomagenesis, we inactivated *Atrx* in *Tp53*-intact and deficient murine neuroepithelial progenitors (mNPCs). We report that Atrx deficiency, particularly when combined with *Tp53* loss, dramatically alters mNPC phenotypes, promoting cellular migration and shifting differentiation markers toward an astrocytic lineage profile. In this way, Atrx loss in mNPCs recapitulates the two biological features most classically associated with diffusely infiltrating astrocytomas. Investigating gene expression changes underlying these phenotypes, we establish compelling correlations with widespread shifts in chromatin accessibility and H3.3 composition, whose positioning strikingly corresponds to the normal distribution pattern of Atrx binding. Finally, we confirm that Atrx-dependent target genes found to mediate migratory and differentiation phenotypes in mNPCs are similarly misexpressed and functionally implicated in *ATRX*-mutant human glioma tumors and cell lines. Taken together, our results demonstrate that ATRX deficiency drives glioma-relevant biology by directly modulating chromatin architecture and composition, thereby influencing the expression of phenotypically crucial gene sets.

## Results

**Atrx loss alters cellular morphology and gene expression**. To recapitulate the cellular and molecular context of ATRX deficiency in human glioma, we extracted primary mNPCs from the brains of mice derived from $Tp53^{-/-}$ and floxed *Atrx* (*Atrx* fl) parental strains, culturing them in stem-like (neurosphere) conditions. Genomics, molecular pathology, and animal modeling studies have repeatedly implicated mNPCs as potential glioma cells of origin[19]. Eliminating *Atrx* expression with cre recombinase-expressing lentivirus (Fig. 1a), we then generated $Tp53^{+/+}$ and $Tp53^{-/-}$ mNPCs harboring either intact or inactivated *Atrx* (hereafter *Atrx+* and *Atrx-*, respectively). Within 2–4 passages, we observed a striking change in morphology restricted to the $Tp53^{-/-}$; *Atrx-* line, whereby cells ceased growing as suspended neurospheres and instead proliferated as an adherent monolayer (Fig. 1b). Acquisition of this phenotype was accompanied by large shifts in gene expression patterns (Fig. 1c). Nevertheless, Atrx deficiency failed to increase proliferation in

either 2-D or 3-D culture (Supplementary Fig. 1a–c), regardless of *Tp53* status. In fact, proliferation was actually reduced in the $Tp53^{+/+}$ context, in accordance with prior studies showing that isolated Atrx deficiency promotes p53-dependent apoptosis[8,20–22]. Further supporting this conjecture, levels of cleaved caspase 3 were elevated in $Tp53^{+/+}$; *Atrx-* mNPCs by western blot (Supplementary Fig. 1d). Consistent with these findings, $Tp53^{-/-}$ mNPCs failed form orthotopic allografts with or without *Atrx* inactivation (Supplementary Fig. 1e). Finally, Atrx deficiency did not induce ALT de novo, as assessed by telomere FISH, in $Tp53^{-/-}$ mNPCs (Supplementary Fig. 1f), recapitulating results from multiple prior reports[23,24].

To better ascertain how the transcriptional programs mobilized by Atrx deficiency reflect glioma biology, we employed single sample gene set enrichment analysis (ssGSEA) to correlate mNPC gene expression patterns with established tumor signatures from TCGA (the Cancer Genome Atlas) and other sources[2,25,26]. We found that Atrx deficiency, in the setting of inactivated *Tp53*, shifted mNPC transcriptional profiles such that they were strongly correlated with the gene expression signatures of several glioma subclasses (Fig. 1d). Neither *Tp53* nor *Atrx* loss alone elicited this response. Particularly robust associations were found with early progenitor-like (EPL), LGG R1, and LGG R3 signatures, all of which are derived from glioma subclasses featuring high rates of *ATRX* mutation[2,4]. Moreover, using published neuroepithelial lineage profiles derived from mice[27], we demonstrated that inactivating *Atrx* in $Tp53^{-/-}$ mNPCs shifted transcriptional patterns toward mature astrocytic and oligodendroglial signatures, at the expense of those corresponding to neurons and oligodendrocyte precursor cells (OPCs; Fig. 1e). These findings indicate that transcriptional alterations induced by ATRX deficiency in cultured mNPCs at least partially recapitulate those occurring in ATRX-mutant gliomas, while also implicating cellular developmental programing in the mediation of this biology.

**Atrx deficiency modulates differentiation state and motility**. To further explore Atrx-dependent transcriptional alterations and their functional consequences, we focused all subsequent analyses on Tp53$^{-/-}$ mNPCs and those genes differentially expressed (fold change>2, *Q*<0.01; ANOVA FDR test) between *Atrx+* and *Atrx−* states (*N* = 2012, Supplementary Fig. 2, Supplementary Data 1). Consistent with our ssGSEA findings, we found that the pool of overexpressed transcripts (*N* = 1105) exhibited significant overlaps with gene ontology (GO) functional sets associated with glial and neuronal development (Fig. 2a). These strong correlations prompted us to investigate the impact of Atrx deficiency on cellular differentiation state and capacity. We examined the expression of established lineage markers in *Atrx+* and *Atrx-* mNPCs in the context of induced differentiation. We found that, over the course of five days in differentiating conditions, Atrx deficiency abrogated the expression of both the neuronal marker tubulin β 3 (Tubb3) and the oligodendrocytic marker Olig2, and instead increased expression of the astrocytic marker glial fibrillary acidic protein (Gfap; Fig. 2b, c). Evaluating other lineage markers yielded similar results (Supplementary Fig. 3). Moreover, Atrx deficiency reduced the number of morphologically mature neurons over the five-day differentiation period in culture (Supplementary Fig. 4), an observation confirmed by Tubb3 immunofluorescence (Fig. 2d, e). We also used our gene expression data to investigate the precise lineage states whose expression patterns best correlated with *Atrx-* mNPCs. Recent work has identified marker-designated subpopulations in the embryonic and neonatal mouse brain corresponding to astrocytic and oligodendrocytic precursors at different developmental stages[28]. Using gene

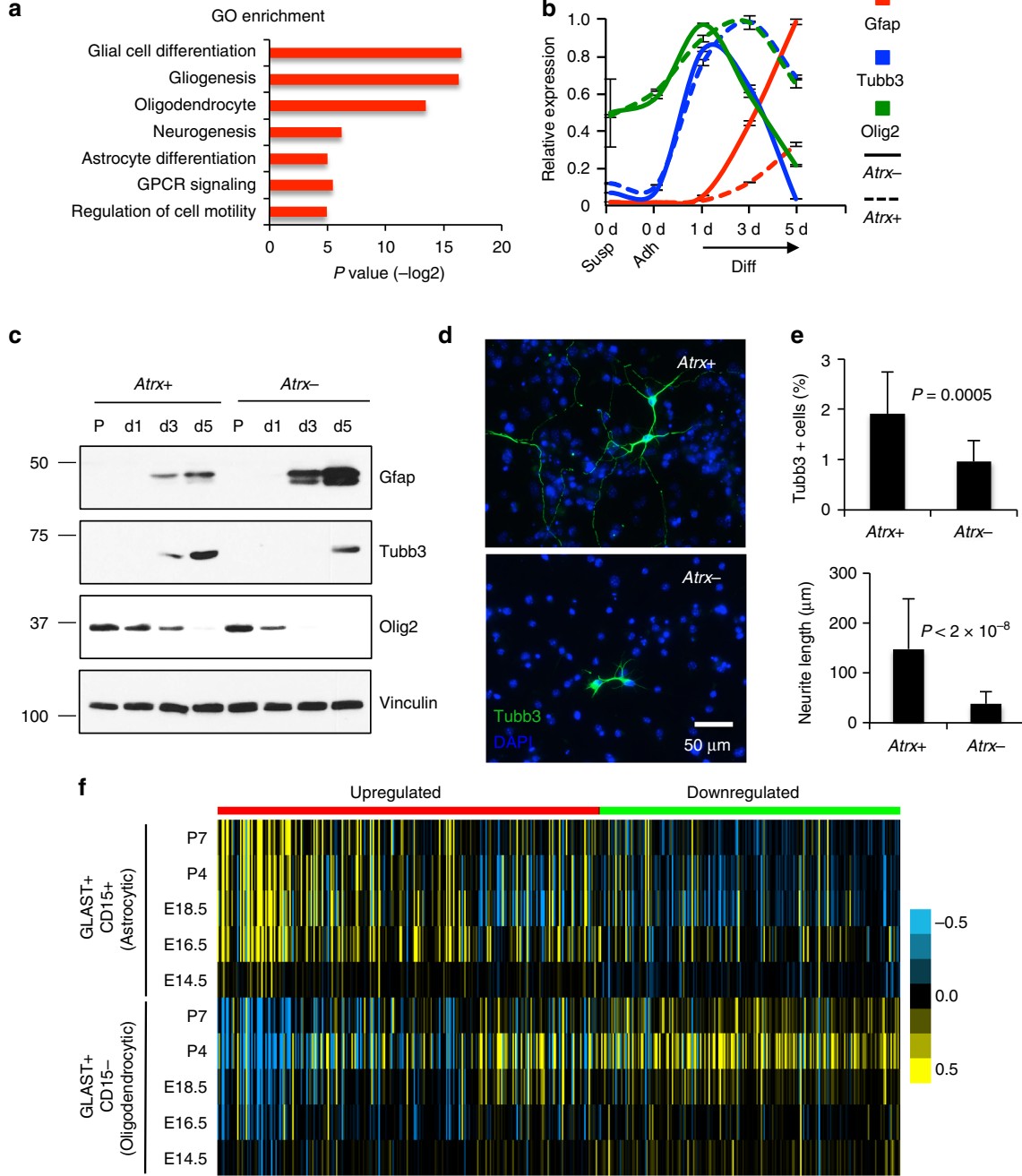

**Fig. 2** Atrx deficiency modulates mNPC differentiation state. **a** Panther analysis of over expressed genes in *Atrx*- mNPCs reveals GO functional associations with development, signal transduction, and cellular motility (GPCR: G protein-coupled receptor). **b** relative transcript levels (RT-qPCR) of Gfap (red), Tubb3 (blue), and Olig2 (green) in *Atrx*+ and *Atrx*- mNPCs (3 replicates per genotype) subjected to differentiation in vitro. Cells in proliferation conditions were sampled in suspension (Susp) and adherent to laminin (Adh), and then in differentiation conditions (Diff) after 1, 3, and 5 days (d). In all cases, data are scaled relative to the highest expression level for each marker across the sample set. Error bars reflect standard deviation. **c** western blot showing levels of histiogenic markers in *Atrx*+ and *Atrx*− mNPCs in proliferation media (P) and in differentiation media for 1, 3, and 5 days (d1, d3, d5; Vinculin loading control). **d**, **e** Tubb3 (green) immunofluorescence images (**d**) and quantification (**e**) showing reduced neuronal processes and a lower number of Tubb3+ cells in Atrx- mNPCs after 5 days in differentiating conditions. In upper plot, data represent the average percentage of Tubb3+ cells counted in 15 fields at ×20 magnification for each genotype; in lower plot, data represent the average length of Tubb3+ protrusions, 30 cells/genotype analyzed (DAPI counterstain; error bars reflect standard deviation; *P* values determined by unpaired two-tailed *t*-test). **f** heat map showing how differentially transcribed genes in *Atrx*- mNPCs (upregulated and downregulated) are expressed in distinct astrocytic and oligodendrocytic precursor compartments. Embryonic days 14.5, E16.5, and E18.5 (E14.5, E16.5, and E18.5) and postnatal days 4 and 7 (P4 and P7) are displayed

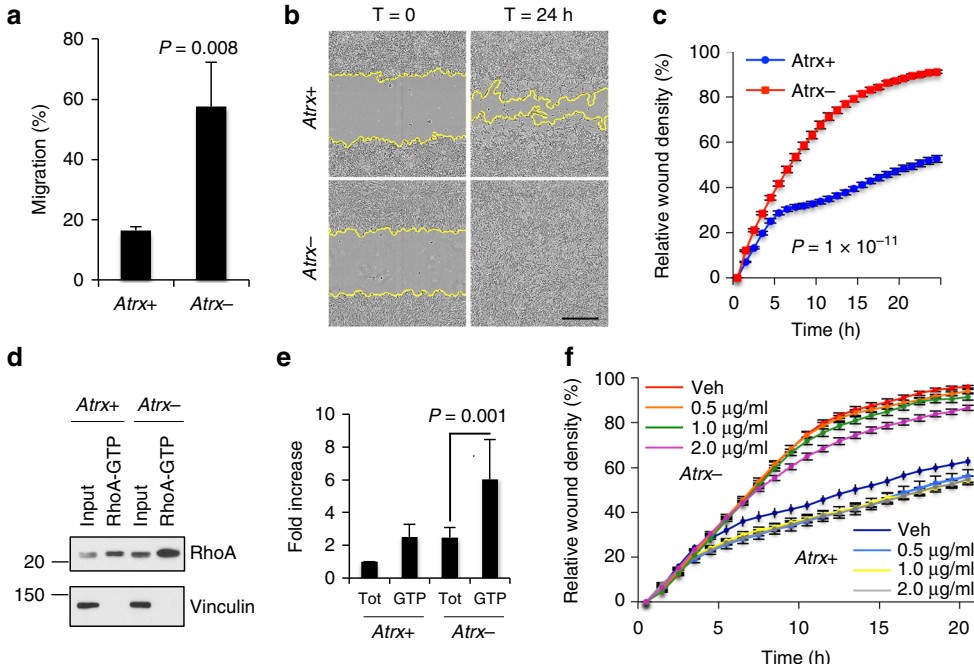

**Fig. 3** Atrx deficiency promotes mNPC motility. Transwell (**a;** 5 replicates) and wound healing (**b, c;** 3 replicates) assays showing increased migratory behavior in *Atrx*- mNPCs (scale bar 300 μm). This phenotype was accompanied by increased GTP-bound RhoA by IP western analysis (**d**) with quantitation by densitometry (**e;** average of 3 biological replicates, Vinculin loading control). **f** wound healing assay for *Atrx*+ and *Atrx*- NPCs treated with the RhoA inhibitor C3 exoenzyme at the designated levels (3 replicates; *Atrx*-: Veh vs 0.5 μg/ml—$P = 0.034$, Veh vs 1 μg/ml —$P = 9.8 \times 10^{-6}$, Veh vs 2 μg/ml —$P = 5.5 \times 10^{-5}$; *Atrx*+: Veh vs 0.5 μg/ml—$P = 8.5 \times 10^{-11}$, Veh vs 1 μg/ml —$P = 2.7 \times 10^{-9}$, Veh vs 2 μg/ml —$P = 1.4 \times 10^{-9}$). *P* values determined by unpaired two-tailed t-test for both transwell assays and western blot densitometry, and by paired two-tailed t-test at all time points for wound healing assays

expression data derived from these distinct cellular compartments, we demonstrated that Atrx-dependent transcriptional alterations were strongly correlated and anti-correlated with those of developing astrocytes and oligodendrocytes, respectively (Fig. 2f). Taken together, these findings indicate that Atrx deficiency alters the differentiation profile of mNPCs, favoring astrocytic over neuronal and oligodendrocytic histiogenesis.

We also found significant correlations between transcripts overexpressed in the setting of Atrx deficiency and gene sets implicated in cellular motility (Fig. 2a). To validate these associations experimentally, we demonstrated that *Atrx*- mNPCs migrated faster than *Atrx*+ isogenics in both transwell and wound healing (scratch) assays (Figs 3a–c and Supplementary Fig. 5). Moreover, IP-western blot showed increased levels of activated, GTP-bound RhoA in *Atrx*- mNPCs (Fig. 3d–e). Rho GTPase signaling has been repeatedly implicated in normal as well as cancer cell migration[29]. Finally, treatment with the RhoA inhibitor C3 exoenzyme, as expected, significantly reduced mNPCs motility in both *Atrx*- and *Atrx*+ contexts, notably with a dose-dependent response only evident in *Atrx*- mNPCs (Fig. 3f). These findings show that Atrx deficiency promotes mNPC migration, at least in part through Rho GTPase-dependent mechanisms. Intriguingly, the increased cellular motility and astrocytic histiogenic profile induced by Atrx deficiency in this in vitro model recapitulate two of the hallmark features associated with diffuse astrocytoma, the predominant *ATRX*-mutant human glioma subtype[30].

**Atrx exhibits a gene regulatory binding pattern in mNPCs.** Having documented widespread transcriptional alterations and disease-relevant phenotypes emerging with Atrx deficiency, we sought to characterize whether and how global epigenomic remodeling might be involved in mediating these effects. Earlier work had shown that ATRX binds to tandem repeat sequences in telomeric regions as well as euchromatin, where its loss can

impact gene expression[18]. To establish the binding distribution of Atrx in mNPCs, we optimized chromatin immunoprecipitation (ChIP) in the murine context, using published sequences where Atrx is known to coordinate. Preliminary studies revealed robust enrichment for Atrx at these predicted binding sites (Fig. 4a). We then proceeded with ChIP-high throughput sequencing (ChIP-seq) in our mNPC lines, using *Atrx*- isogenics as negative controls. Our findings (Fig. 4b–e, Supplementary Data 2, 3) contrasted sharply with what had previously been reported in mESCs[18], despite an almost identical degree of overlap (66% versus 65%) between Atrx enrichment peaks and tandem repeat sites in the murine genome. In mNPCs, Atrx exhibited a much larger number of binding sites (73,723 and 76,164 in $Tp53^{-/-}$ and $Tp53^{+/+}$ lines, respectively) than in mESCs (19,103). Notable qualitative differences were also apparent, with only 8.0% overlap between mESC and $Tp53^{-/-}$ mNPC peaks. By contrast, enrichment peaks for $Tp53^{+/+}$ and $Tp53^{-/-}$ mNPCs overlapped extensively (83.2%), indicating that p53 loss has, at best, only a modest effect on the genomic positioning of Atrx.

The distribution of Atrx binding with respect to open reading frames (ORFs) was also quite different in NPCs relative to mESCs. Whereas Atrx binding in mESCs was largely restricted to intergenic sites, the majority of Atrx peaks in mNPCs (67%) involved either gene bodies or promoter regions (Fig. 4b). Moreover, heatmap analysis demonstrated that Atrx positioning strongly correlated with both transcriptional start sites (TSSs) and enhancer regions in mNPCs, associations that were inapparent in mESCs (Fig. 4c, d). These findings are consistent with Atrx assuming a more gene regulatory role in mNPCs than in mESCs. Further supporting this notion, Atrx-bound genes—defined as those harboring one or more Atrx binding peaks within 10 kb of their TSSs (Supplementary Data 1)—were strongly enriched for transcripts differentially expressed in the context of Atrx deficiency ($P = 4.5 \times 10^{-149}$; hypergeometric test).

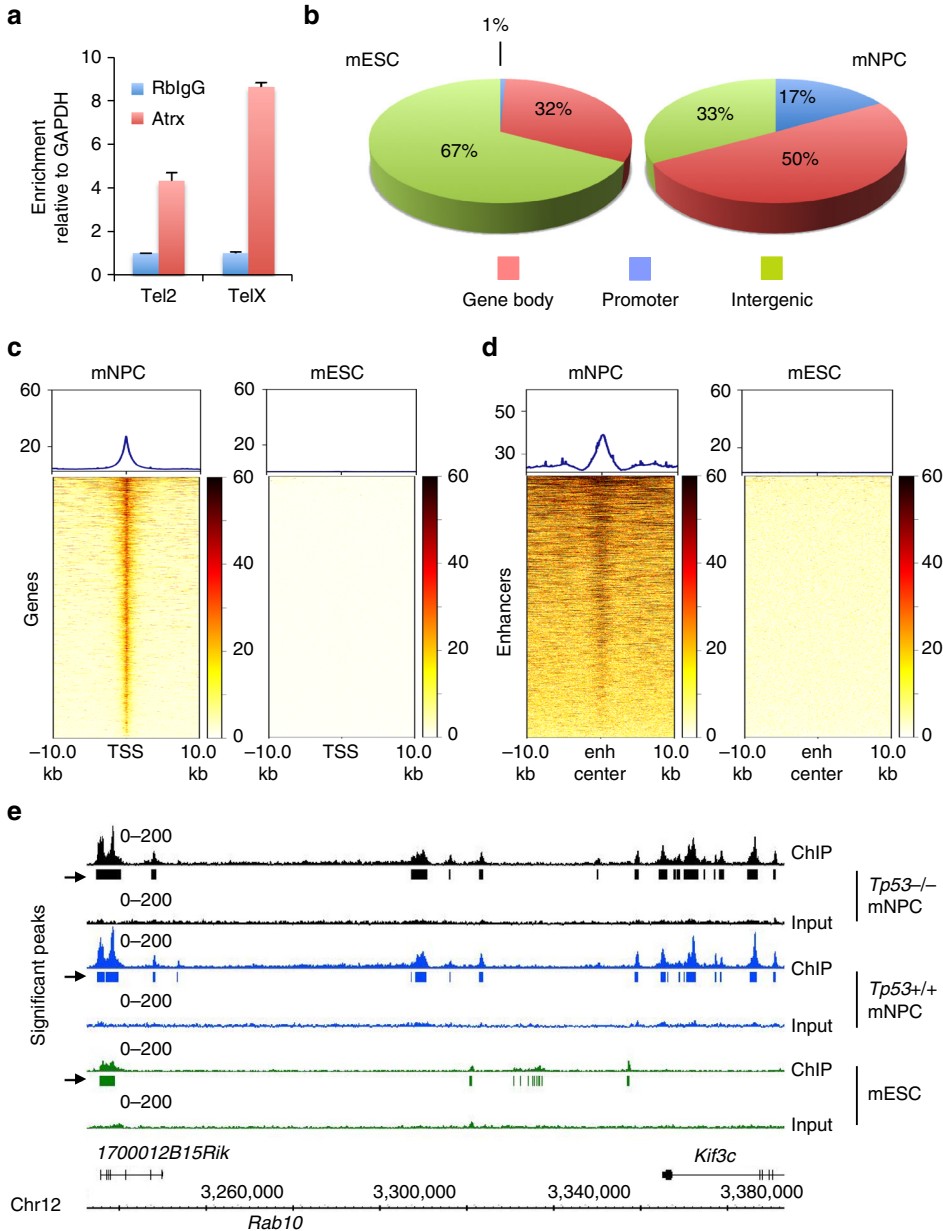

**Fig. 4** Atrx exhibits an extensive, gene regulatory binding pattern in mNPCs. **a** qPCR analysis of Atrx ChIP in mNPCs shows enrichment over rabbit immunoglobulin control (RbIgG) at previously reported Atrx binding sites (Tel2: $P = 0.0010$, TelX: $P = 3.8 \times 10^{-8}$; 3 replicates; error bars reflect standard deviation; $P$ values determined by unpaired two-tailed $t$-test). **b–d** Atrx ChIP-seq in mNPCs reveals greater numbers of enrichment peaks within gene body and promoter regions (**b**) than previously reported in mESCs[18]. Notable clustering around TSSs (**c**) and enhancers (**d;** enh center) is also seen, contrasting sharply with Atrx peaks in mESCs. **e** sample ChIP-seq traces for mNPCs ($Tp53^{-/-}$ and $Tp53^{+/+}$) and mESCs showing distinct Atrx binding patterns between cell types

**Atrx deficiency alters chromatin accessibility genome-wide.** Given the extensive and gene-associated distribution of Atrx in mNPCs, we hypothesized that its deficiency would promote widespread, transcriptionally relevant shifts in chromatin state. To explore this possibility, we performed assay for transposase-accessible chromatin with high-throughput sequencing (ATAC-seq)[31] in both $Atrx+$ and $Atrx-$ mNPCs, identifying genomic loci whose accessibility were significantly altered in the setting of Atrx deficiency. This analysis revealed thousands of sites across the genome where chromatin exhibited either significantly increased ($N = 11607$) or decreased ($N = 7887$) accessibility upon $Atrx$ inactivation (Fig. 5a, Supplementary Data 4, 5). We subsequently referred to these sites as "open" and "closed", respectively.

Remarkably, 82.3% of ATAC-seq-altered sites (80.4% for open and 84.8% for closed) exhibited overlap with the genomic coordinates of Atrx enrichment peaks (Fig. 5b). These findings indicate that altered chromatin accessibility arising in the context of Atrx deficiency is largely restricted to the immediate vicinity of Atrx binding sites. Not surprisingly, we also documented genomic associations between ATAC-seq-altered sites and both TSSs and enhancer regions, reflecting their intimate relationship with Atrx binding loci (Fig. 5c).

We then examined whether these shifts in chromatin accessibility were associated with differential gene expression, focusing on ORFs harboring ATAC-seq open and/or closed sites within 10 kb of their TSSs (Supplementary Data 1). This analysis

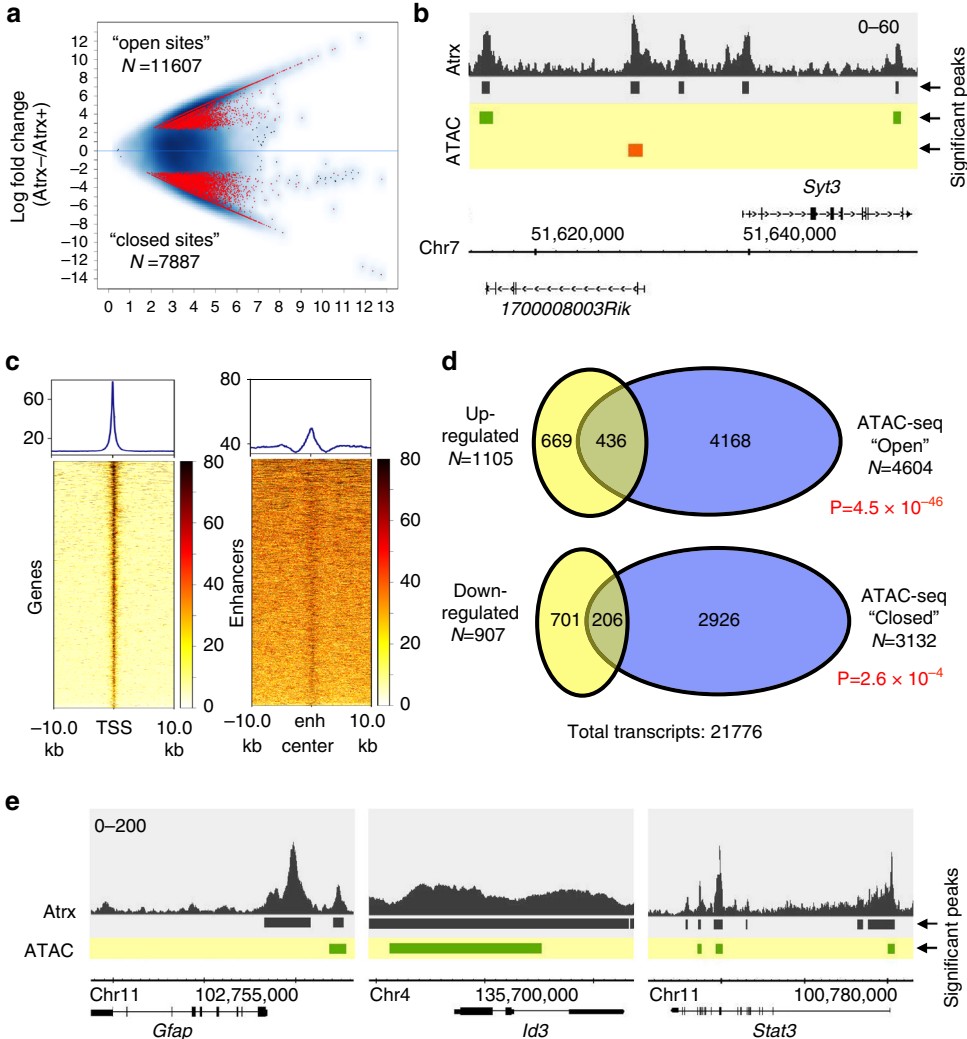

**Fig. 5** Atrx deficiency alters chromatin accessibility genome-wide at vacant Atrx binding sites. **a** Volcano plot showing differentially accessible genomic sites (open or closed) arising with Atrx deficiency in mNPCs. **b** Sample trace showing the extent of overlap between Atrx ChIP-seq peaks (black) and ATAC-seq open (green) and closed (red) regions. **c** heat maps showing the spatial relationships between TSSs, enhancers (enh center), and differentially accessible regions arising in the context of Atrx deficiency. **d** Venn diagrams demonstrating significant overlaps between upregulated genes and genes approximating ATAC-open sites, and downregulated genes and genes approximating ATAC-closed sites (±10 kb; P values determined by hypergeometric test). **e** sample traces of astrocytic lineage markers or regulators showing their genomic associations with Atrx binding sites (black) and ATAC-seq open regions (green)

revealed highly significant correlations between open sites and upregulated genes, and between closed sites and downregulated genes (Fig. 5d). The converse relationships were not observed. Reassuringly, key genes associated with astrocytic lineage (*Gfap*, *Id3*, and *Stat3*)[32–34] and upregulated in the context of the differentiation phenotype described above were also associated with ATAC-seq open sites (Fig. 5e, Supplementary Data 1). These findings reveal intriguing links between Atrx deficiency, chromatin structure, and gene expression. Moreover, they suggest that the majority of transcriptionally relevant epigenetic events occurring with Atrx deficiency derive from locally dysregulated chromatin composition at vacant Atrx binding loci.

**Gna13 promotes Atrx-deficient glioma cell motility.** Given the association of ATAC-seq open sites with upregulated astrocytic lineage markers (see above), we hypothesized that regions of increased chromatin accessibility would also arise in spatial proximity to genes mediating the observed motility phenotype of *Atrx*- mNPCs. Accordingly, we cross-referenced genes harboring

ATAC-seq open sites and Atrx enrichment peaks within 10 kb of their TSSs with upregulated transcripts also bearing functional associations with motility, migration, and/or invasion by either ingenuity pathway analysis (IPA) or GO (Fig. 6a). This process identified 38 genes (Supplementary Table 1), which we then subjected to systematic siRNA knockdown in the context of the transwell migration assay. We found several genes for which significant repression reproducibly correlated with abrogation of cell migration (Fig. 6b). Among these targets, knockdown of *Gna13*, which encodes the α subunit of the heterotrimeric G-protein Gα13, had the greatest effect. Several studies have linked the G12 G-protein subfamily, which includes Gα12 and Gα13, to cancer cell proliferation and motility[35]. Moreover, G12 G-proteins signal extensively through downstream RhoA GTPases[35], which our earlier findings had already implicated in Atrx-deficient mNPC motility (Fig. 3d–f).

To assess whether *GNA13* might facilitate similar processes in ATRX-deficient human disease, we examined TCGA gene expression data for lower-grade glioma (LGG), focusing our

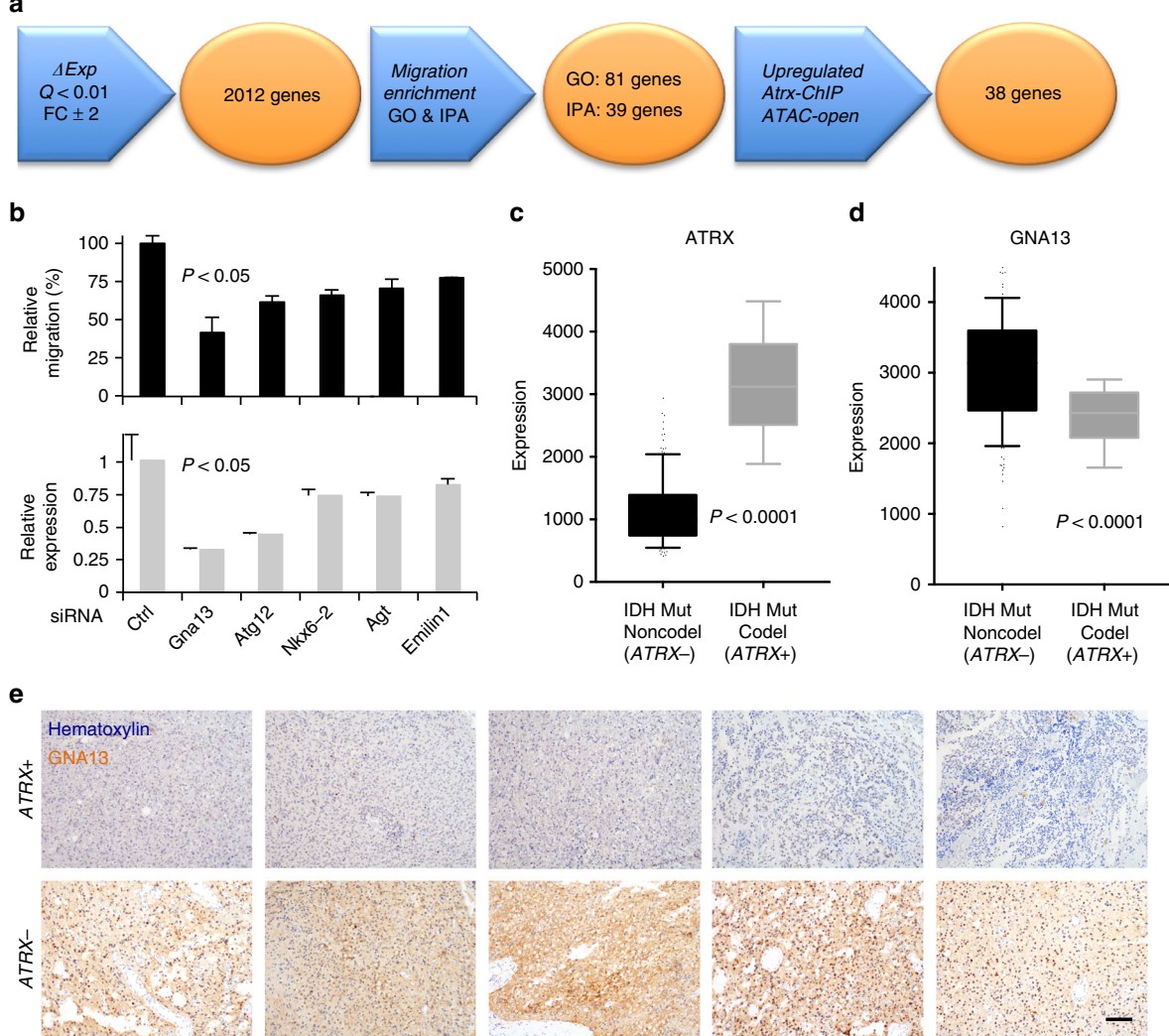

**Fig. 6** Gna13 promotes cell migration in *Atrx*- mNPCs and is upregulated in ATRX-deficient gliomas. **a** schematic demonstrating strategy to identify upregulated genes promoting mNPC motility in the setting of Atrx deficiency (FC: fold change). **b** relative mNPC transwell migration (upper chart) following siRNA knockdown (25 nM) of the indicated gene target (lower panel), as confirmed by RT-qPCR (error bars reflect standard deviation; *P* values determined by unpaired two-tailed *t*-test). **c**, **d** LGG expression data from TCGA showing levels of *ATRX* and *GNA13* expression in ATRX-intact (*ATRX*+) and ATRX-deficient (*ATRX*-) gliomas. Error bars reflect the 10–90 percentile spread and *P* values determined by unpaired two-tailed *t*-test. **e** micrographs of *ATRX*+ and *ATRX*- gliomas showing differential GNA13 expression by IHC (hematoxylin counterstain; scale bar 100 μm)

analysis on those tumors harboring an *IDH1* or *IDH2* mutation. As indicated above, *ATRX* mutation in adult glioma arises almost exclusively in the context of concurrent IDH mutation. Moreover, recent work has shown that IDH-mutant gliomas are composed of two predominant subtypes, those with and those without 1p/19q chromosomal codeletion (IDHmut-codel and IDHmut-noncodel), with the IDHmut-noncodel subclass not only harboring the vast majority of *ATRX*-mutant tumors, but also exhibiting uniformly low-level *ATRX* expression[2]. Our analysis confirmed this latter association, while also demonstrating significantly higher levels of *GNA13* transcript in IDHmut-noncodel gliomas (Fig. 6c, d). Notably, two additional motility regulators, *AGT* and *EMILIN1*, implicated in our siRNA screen were similarly overexpressed as were the astrocytic markers/regulators *GFAP*, *ID3*, and *STAT3* (Fig. 6b and Supplementary Fig. 6). In parallel, we performed immunohistochemical studies on ten IDH-mutant gliomas with known *ATRX* expression status. We found that GNA13 levels were consistently higher in *ATRX*- tumors than in *ATRX*+ counterparts (Fig. 6e).

We then evaluated the expression of GNA13 protein in primary glioma stem cell (GSC) lines derived from patient tumors. GS 5-22, an *IDH1*-mutant (IDH1 R132H) GSC also featuring *ATRX* mutation, exhibited higher *GNA13* expression than TS 603, an *IDH1*-mutant, *ATRX*-wild type line, as well as two IDH-wild type GSCs (GS 600 and TS 543; Fig. 7a, b). Moreover, GS 5-22 exhibited increased transwell migration relative to TS 603 (Fig. 7c). These relationships were recapitulated in a second *IDH1*-mutant, *ATRX*-mutant GSC line (GS 8-18) and elevated levels of *GNA13* transcript were also seen in third *IDH1*-mutant *ATRX*-mutant GSC line (JHH-273) only capable of forming subcutaneous xenografts[36] (Supplementary Fig. 7). Finally, we used electroporation to transiently restore human *ATRX* expression in both mNPCs and GS 5-22 cells. While our protocol resulted in only ~20–30% transfection efficiency, we were nevertheless able to demonstrate significant reversion of the migratory phenotype accompanied by a modest, but significant decrease in *Gna13/GNA13* expression in both cell lines (Fig. 7d–j).

Taken together, these data firmly implicate GNA13 and

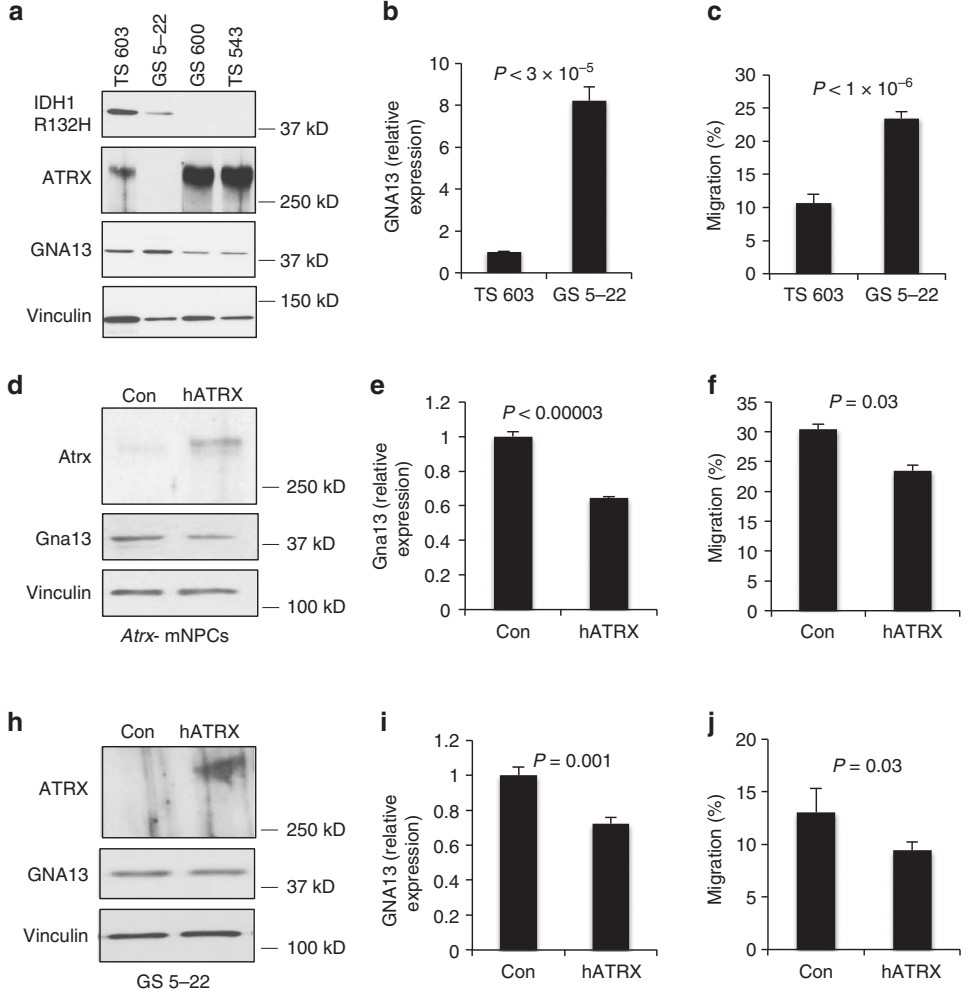

**Fig. 7** GNA13 promotes cell migration in *ATRX*-mutant GSCs. **a**, **b** western blot (**a**) and RT-qPCR (**b**; three replicates) showing increased *GNA13* expression in *IDH1*-mutant (IDH1 R132H), ATRX-deficient GSC line (GS 5–22). **c** transwell migration assay (5 replicates) demonstrating increased motility in *IDH1*-mutant, ATRX-deficient GSC line (GS 5-22) relative to *IDH1*-mutant, ATRX-intact GSC line (TS 603). **d**–**e** western blot (**d**) and RT-qPCR (**e**; three replicates) showing decreased *Gna13* expression upon transient transfection of Atrx- mNPCs with human *ATRX* (hATRX) relative to control vector (Con). **f** transwell migration assay (5 replicates) demonstrating decreased motility in hATRX transfected, *Atrx*- mNPCs. **h**–**i** western blot (**h**) and RT-qPCR (**i**; three replicates) showing decreased *GNA13* expression upon transient transfection of GS 5-22 cells with hATRX relative to Con. **j** transwell migration assay (5 replicates) demonstrating decreased motility in hATRX transfected GS 5-22 cells. All western blots used Vinculin as a loading control. In all cases, error bars reflect standard deviation and *P* values determined by unpaired two-tailed *t*-test

downstream RhoA signaling, along with potentially other ATRX-responsive genes, in the process of ATRX-dependent glioma cell migration. In a larger sense, they also suggest that functionally relevant epigenetic and transcriptional alterations mobilized by Atrx deficiency in murine mNPCs are also operative in *ATRX*-mutant human gliomas.

**Atrx-dependent epigenomic shifts correlate with H3.3.** ATRX has been shown to regulate incorporation of H3.3 histone monomers into nucleosomes[16,17]. To determine whether ATRX-dependent shifts in chromatin accessibility involve altered H3.3 composition, we performed H3.3 ChIP in *Atrx*+ and *Atrx*-mNPCs, focusing on genomic loci previously demonstrated to be both Atrx-bound and associated with differentially expressed genes in the context of Atrx deficiency. We found significant changes in H3.3 enrichment within these regions, which included binding peaks approximating several phenotypically relevant transcripts, such as *Gna13* and *Gfap* (Fig. 8a, b and Supplementary Fig. 8a–8d). By contrast, sampled genes not exhibiting differential expression with Atrx deficiency that were either Atrx-

bound (*Tchp*) or unbound (*Cd163*) failed to exhibit significant H3.3 enrichment over IgG control (Supplementary Fig. 8e, f). We also performed H3.3 ChIP-seq in both *Atrx*+ and *Atrx*- mNPCs to ascertain the effects of Atrx deficiency on H3.3 composition genome-wide. While this global analysis lacked the sensitivity of our focused, RT-qPCR-based approach, we were still able to identify 560 sites of differential H3.3 incorporation and demonstrate 77.2% overlap with Atrx binding peaks (Fig. 8c, Supplementary Data 6). Finally, focused ChIP analysis in *ATRX*+ (TS 603) and *ATRX*- (GS 5-22) GSCs revealed differential H3.3 incorporation in association with *GNA13* (Fig. 8d), extending our findings into bona fide human disease models. These results indicate that Atrx deficiency abnormally alters H3.3 composition at vacant Atrx binding sites, which in turn mediates, at least in part, shifts in chromatin accessibility and gene expression driving glioma-relevant phenotypes.

**Discussion**

*ATRX* inactivation defines large subsets of adult and pediatric glioma[2–5], where its sheer frequency argues for a major role in

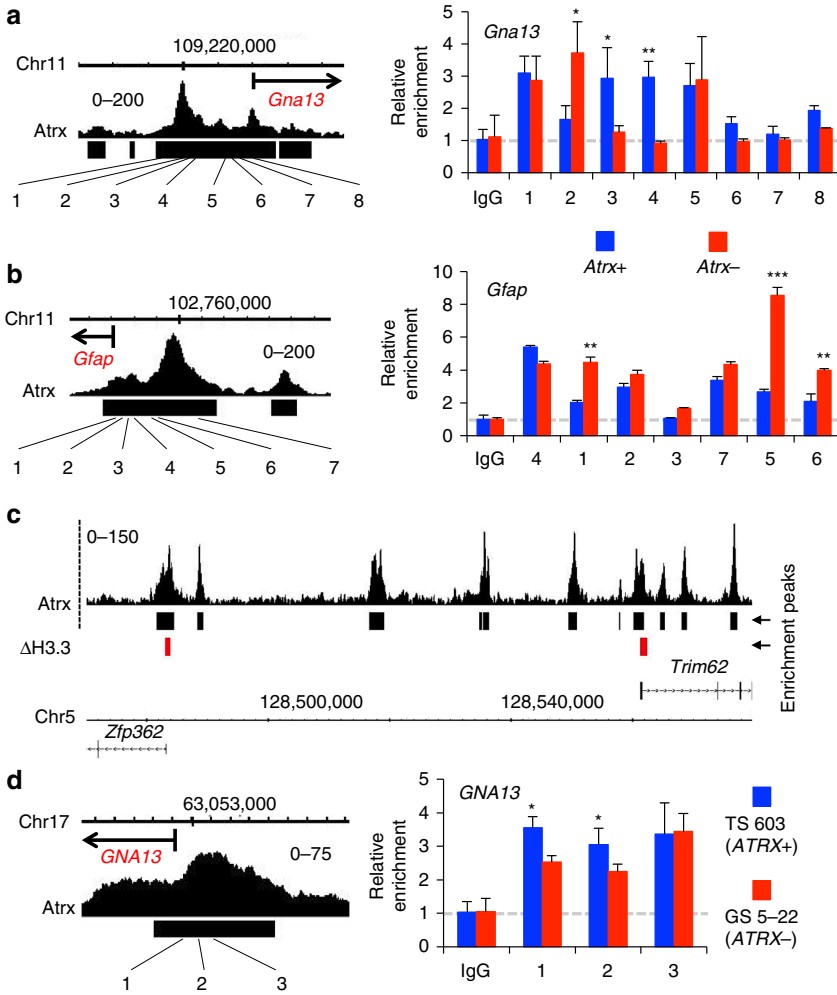

**Fig. 8** Atrx-dependent epigenetic and transcriptional alterations correlate with H3.3 composition. **a**, **b** H3.3 ChIP qPCR from Atrx-bound genomic loci approximating *Gna13* and *Gfap* reveals differential enrichment between *Atrx+* and *Atrx-* mNPCs at many points. Positioning of loci subjected to qPCR relative to Atrx binding peaks and open reading frames (*Gna13* and *Gfap*) is also shown (left panels). **c** sample trace showing relationship of Atrx ChIP-seq peaks (black) to sites of significantly differential H3.3 composition (ΔH3.3) arising with Atrx deficiency (red). **d** H3.3 ChIP qPCR from Atrx-bound genomic loci approximating *GNA13* reveals differential enrichment between *ATRX+* (TS 603) and *ATRX-* (GS 5-22) human GSCs (right panels; *P < 0.05; **P < 0.001; ***P < 0.0001. In all cases, error bars reflect standard deviation and P values determined by unpaired two-tailed *t*-test

oncogenesis. Considering the established role of ATRX as an epigenetic regulator, we hypothesized that its inactivation induces glioma-relevant phenotypes by way of chromatin remodeling and downstream shifts in gene expression. To investigate this possibility, we re-engineered the basic molecular features of ATRX-deficient glioma in a disease-relevant cellular context. Primary GSCs harboring *ATRX* mutation have thus far proven difficult to culture and genetically manipulate, and the one Atrx-deficient murine model generated to date layers *Atrx* loss upon an already high-grade, NRAS-driven glioma[37], affording limited opportunity for the study of more subtle phenotypes. As indicated above, a number of investigations have implicated mNPCs as potential glioma cells of origin[19]. Although no in vitro system can perfectly recapitulate the process of oncogenesis, these primary cells, when not excessively passaged, offered an opportunity to study the gliomagenic effects of Atrx deficiency in a molecular background not extensively altered by other cancer-promoting physiological dysruptions.

The sufficiency of *ATRX* inactivation, either alone or in combination with *TP53* loss, to induce proliferative gliomagenesis has yet to be demonstrated. Our work, along with that described in another recent report[38], supports the notion that the oncogenic

effects of ATRX deficiency are more measured and may be dependent on as yet uncharacterized molecular factors. Nevertheless, *Atrx* inactivation induced widespread gene expression changes in mNPCs, especially when paired with *Tp53* loss, recapitulating its core mutational context in human gliomas. Prior functional analysis of ATRX deficiency in the central nervous system has occurred primarily in the *TP53*-intact setting[8,21,22], which tends to result in cell death and, as such, may not represent an ideal model for glioma biology. Indeed, we also observed increased cell death in Atrx-intact mNPC isogenics. Reassuringly, combined *Atrx* and *Tp53* loss altered mNPC transcriptional patterns such that they strongly correlated with several glioma signatures, particularly those derived from ATRX-deficient tumors.

Atrx-dependent gene expression changes in mNPCs were accompanied by altered cellular morphology, increased motility, and a shift in differentiation markers toward an astrocytic histogenic profile. These latter two phenotypes recapitulate two cardinal features of *ATRX*-mutant adult and pediatric gliomas, namely widespread tumor cell invasion into surrounding brain and astrocytic histopathology[30]. Until quite recently, primary brain tumors were classified solely on the basis of presumed

histiogenesis[39]. Within this schema, the morphological and immunohistochemical features of diffuse astrocytoma were thought reflect fundamental derivation from an astrocytic precursor, although direct evidence supporting this hypothesis was limited[40]. By demonstrating that ATRX deficiency directly mobilizes the expression of key astrocytic markers and regulators, we provide compelling evidence that this disease-defining molecular alteration, in and of itself, directs precursor cells of unspecified lineage toward an astrocytic phenotype at the expense of normal neuronal and/or oligodendrocytic differentiation. These findings support the notion that apparent lineage derivation in cancer is a fluid concept, dependent on molecular as well as cellular context, while also suggesting that mobilization of latent developmentally relevant pathways may underlie a significant subset of ATRX-deficient biology.

The migratory behavior of diffusely infiltrating gliomas, including *ATRX*-mutant astrocytomas, represents a major determinant of their malignant potential, effectively rendering them incurable by surgical resection. Accordingly, an improved understanding of the molecular events driving this phenotype would have tangible therapeutic implications. Our analysis implicated multiple Atrx target genes in glioma cell motility, including *Gna13*, whose direct transcriptional upregulation and downstream signaling through RhoA GTPases promoted mNPC migration. Moreover, our analysis of human tissue samples and multiple primary GSCs revealed that this molecular mechanism might also be operative in ATRX-deficient gliomas. In this way, we established compelling mechanistic links between ATRX deficiency and targetable molecular networks classically implicated in cancer cell motility. Prior work using LN 229 glioblastoma cells as a model system had found reduced cell migration arising with ATRX deficiency[41]. However, a more recent report has validated our findings in neural stem cells also harboring molecular alterations (*IDH1* mutation and *TP53* inactivation) characterizing ATRX-deficient astrocytoma[38].

To better understand how Atrx, or lack thereof, modulates disease-relevant transcriptional profiles in murine mNPCs, we assessed its genomic distribution by ChIP-seq. We found an unexpectedly extensive binding pattern that contrasted sharply, both quantitatively and qualitatively, with what had been previously reported for mESCs[18]. Although the different number of Atrx binding peaks identified in our mNPC profiles may partially reflect technical variability between the two studies, the notable correlation of Atrx distribution with gene bodies and promoter regions, not seen in mESCs, speaks to a hitherto unappreciated, cell line-specific functionality for Atrx. In particular, a limited set of histiogenic lineages appear to feature epigenomic states unusually dependent on Atrx for normal gene regulation, and by extension, exquisitely sensitive to *Atrx* inactivation. Intriguingly, they also reflect the relatively narrow spectrum of tumor types exhibiting high rates of *ATRX* mutation, which are notably enriched for neuroepithelial derivation (e.g., adult and pediatric glioma, neuroblastoma, and pancreatic neuroendocrine tumor)[6]. In this way, the oncogenic effects of ATRX deficiency likely require specific cellular contexts to achieve their full expression.

The molecular context permitting ATRX-deficient gliomagenesis may be similarly restricted. Indeed, for reasons described above, *ATRX*-mutant gliomas almost invariably harbor *TP53* mutations. Moreover, the vast majority of adult and pediatric gliomas with ATRX deficiency also contain mutant IDH and H3.3 genes, respectively, both of which have been shown to significantly and distinctively impact global epigenomic landscapes[42–44]. In the present study, we chose to focus our analysis on the core signature of *ATRX* and *TP53* inactivation, both to identify the precise epigenetic and transcriptional consequences of ATRX deficiency and also obtain findings broadly generalizable

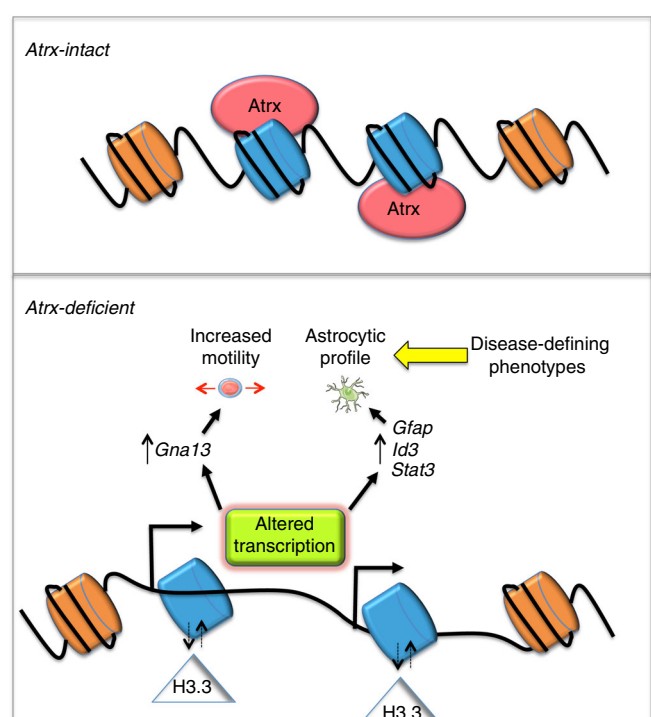

**Fig. 9** Model for epigenomic and transcriptional dysregulation occurring with ATRX deficiency. ATRX inactivation alters chromatin structure and accessibility in the immediate vicinity of vacant ATRX binding sites (blue), in part due to shifts in the incorporation of H3.3. These changes induce the misexpression of locally situated genes, promoting the acquisition of disease-defining cellular phenotypes, such as motility and altered differentiation

across adult and pediatric glioma. We acknowledge, however, that including either mutant IDH or H3.3 proteins in our mNPC system could promote additional epigenomic modifications, impacting transcriptional and phenotypic profiles. That being said, our validation studies in ATRX-deficient human gliomas suggest that the cell migratory and differentiation phenotypes identified in our mNPC models are driven primarily by *ATRX* loss.

Atrx inactivation in mNPCs had profound effects on the global epigenome. We observed widespread changes in chromatin accessibility whose positioning strikingly reflected both the normal binding distribution of Atrx as well as Atrx-dependent transcriptional alterations. These data support a model in which *ATRX* loss modulates chromatin structure primarily in the immediate vicinity of its vacant binding sites, dysregulating local gene expression and inducing glioma-relevant phenotypes (Fig. 9). These localized shifts in chromatin architecture imply underlying alterations in nucleosome composition. Supporting this conjecture, and consistent with prior reports[16,17], we found altered levels of H3.3 enrichment in *Atrx+* and *Atrx-* isogenic mNPCs by both focused and global ChIP analysis at gene-associated Atrx-bound loci whose differential expression mediated key Atrx-deficient phenotypes. Findings related to *GNA13* were further validated in human GSCs, albeit in a non-isogenic context. Prior studies have shown that H3.3 compositional changes occur during neuroepithelial differentiation, and that ATRX mediates H3.3 deposition at telomeres as well as heterochromatin sites associated with the silenced alleles of imprinted genes[15,45]. Our findings expand on these conclusions, demonstrating a novel, cell type-specific role for ATRX in nervous system development grounded in broad H3.3 regulation at ORFs

genome-wide. Moreover, our documentation of genomic sites exhibiting either increased or decreased H3.3 content in the setting of Atrx deficiency suggests that Atrx may play a more general role in precisely titrating H3.3 composition at target site nucleosomes, rather than simply promoting H3.3 incorporation.

Recent work has demonstrated that ATRX deficiency also modulates the composition of macroH2A histones in chromatin as well as the levels of H3K27 and H3K9 trimethylation marks[12–15,46]. Although our data do not exclude the involvement of these regulatory mechanisms, we note that none of these prior studies localized the effects of ATRX deficiency so precisely to vacant ATRX binding sites. Accordingly, our findings support a distinct mechanism of ATRX-based transcriptional control at least partially dependent on differential H3.3 incorporation into gene-associated nucleosome complexes. Moreover, they establish firm links between ATRX-mediated H3.3 regulation and functionally relevant gene expression programs on a global scale.

Prior reports have revealed compelling associations between ATRX deficiency and ALT[9,10,37,47], a process that likely enables cellular immortality in ATRX-mutant tumors. That virtually all IDH-mutant adult gliomas feature either ATRX deficiency or activating mutations in TERT[2], which encodes the catalytic core of telomerase, speaks to the absolute necessity of pathological telomere maintenance across the tumor group. However, only a small fraction of IDH-mutant, TERT-mutant tumors do not also feature 1p/19q codeletion[2]. Accordingly, this same genetic evidence implies that ATRX deficiency must induce other essential phenotypes in affected tumors. Moreover, while ATRX mutations are frequently found in H3.3 mutant pediatric gliomas, TERT mutations are rare[48], further supporting the notion that the oncogenic sequelae of ATRX deficiency are not limited to ALT. Our data support this conjecture by demonstrating specific, ALT-independent molecular mechanisms mediating disease-relevant biological behavior. That Atrx inactivation was insufficient to induce a full ALT phenotype in our mNPCs is consistent with earlier studies in both murine and human experimental systems[23,24,37]. These findings illustrate the multifactorial influences of ATRX deficiency on the oncogenic process.

In summary, we describe, for the first time, the epigenomic consequences of ATRX deficiency in putative glioma cells of origin, along with their effects on gene expression programs and cellular differentiation and motility phenotypes. In doing so, our work characterizes targetable molecular pathways mediating well-established malignant behavior in diffusely infiltrating astrocytic gliomas, while also revealing discrete examples of how mutational disruption of chromatin regulation impacts the expression of tumor-specific biology.

## Methods

**Mice and genotyping.** All animal experiments were performed in accordance with protocols approved by the Memorial Sloan-Kettering Cancer Center (MSKCC) and MD Anderson Cancer Center (MDACC) Institutional Animal Care and Use Committees (protocol numbers 09-09-017 and 00001597-RN00). Tp53$^{-/-}$ (B6.129S2-Trp53tm1Tyj/J; Jackson Laboratory) and Atrx fl mice[8] were crossed to obtain different genotypic combinations. Pups were genotyped by PCR from tail-derived genomic DNA (see Supplementary Data 7 for oligonucleotides). For Tp53, wild type and knockout bands are 375 bp and 525 bp, respectively[49]. For Atrx fl, wild type and floxed bands are 1.0 kb and 1.5 kb, respectively.

**mNPC isolation and culture.** Forebrains were removed from p1 mouse pups and mechanically homogenized by repetitive pipetting in NeuroCult Basal Medium containing NeuroCult Proliferation Supplement, 20 ng/ml EGF, 10 ng/ml basic FGF, 2 µg/ml heparin (Stemcell Technologies), 50 units/ml penicillin, and 50 µg/ml streptomycin (Thermo Fisher Scientific). Homogenized samples were then cultured in a humidified incubator at 5% $CO_2$ and 37 °C. To inactivate Atrx, cells were infected with a Cre-IRES-Puro lentiviral plasmid (a gift from Darrell Kotton; Addgene plasmid #30205)[50] and selected with 1 µg/mL of puromycin (Sigma-Aldrich). To induce differentiation, cells were plated on dishes coated with 10 µg/ml laminin (Sigma-Aldrich) and cultured in NeuroCult Basal Medium containing NeuroCult Differentiation Supplement (Stemcell Technologies). For experiments, cells were used at passages 5–10.

**GSC culture.** Human GSC lines were cultured in DMEM/F12 plus B-27 supplement (Gibco), 20 ng/ml EGF, 10 ng/ml basic FGF, 2 µg/ml heparin, 50 units/ml penicillin, and 50 µg/ml streptomycin. SNP-based authentication and mycoplasma testing occurred every 3 months in culture

**Transfections.** Human ATRX was re-expressed in mNPCs and GSCs by electroporation using the plasmid IF-GFP-ATRX (gift from Michael Dyer: Addgene # 45444)[51]. Electroporations were performed using Nucleofector™ Kit for Mouse Neural Stem Cells (Lonza) according to manufacturer's guidelines. 20 µg of either IF-GFP-ATRX (hATRX) and control pGFP (Con) plasmid were transfected in 5 × 10⁶ cells, which were harvested 48–72 h later for RNA and protein extraction and/or migration assay.

**Proliferation assays.** Cell proliferation was assessed by multiple methods. (1) 2D proliferation assay: $5 \times 10^3$ mouse mNPCs were grown in triplicate on laminin-coated (10 µg/ml) 96-well plates and stopped at different time points. Proliferation was evaluated using the CellTiter-Glo Luminescence Cell Viability Assay (Promega), following manufacturer instructions. (2) 3D-proliferation assay: mNPCs in triplicate wells were embedded in 0.5% 2-hydroxyethyl agarose (Agarose VII, Sigma-Aldrich) and cultured for 1 week. Colonies were then stained with crystal violet and counted. (3) Cell Cycle analysis: cell cycle profiles were analyzed by propidium iodide staining (50 µg/ml propidium iodide and 100 µg/ml RNase A) followed by fluorescence-activated cell sorting (FACS) analysis. (4) In vivo growth of mNPCs was assessed by intracranial injection of $1 \times 10^5$ luciferase expressing cells in 8-week old Nu/Nu mice (Charles River). Growth was monitored by Xenogen IVIS imaging following luciferin injection and data were analyzed by IVIS Lumina LivingImage software.

**Cell motility assays.** To assess transwell migration, cells were labeled with 5 µg/ml of the green fluorescent tracer DiO (Life Technologies). A total of $2 \times 10^5$ cells were added to the upper compartment of a 24-well FluoroBlok insert (Corning) coated with laminin 20 µg/ml, in NeuroCult Basal Medium plus 0.1% BSA (Sigma-Aldrich). 20 ng/ml EGF was used as a chemoattractant in the lower chamber. After 24 h, invading cells were detected with a Cytation 3 Imaging System microplate reader (BioTek). Displayed data reflects 5 biological replicates. For wound-healing assays, exponentially growing cells were seeded ($1 \times 10^5$) in a 96-well laminin-coated ImageLock plate (Essen BioScience) in 3 biological replicates to create a dense monolayer and then scratched with the Wound Maker (Essen BioScience), following manufacturer instructions. NeuroCult Basal Medium plus 0.1% BSA and 20 ng/ml EGF was added after washing with PBS, and wound closure was monitored by an IncuCyte Zoom System (Essen BioSciences) for 24 h. Images were analyzed with Incucyte Scratch Wound Cell Migration Software (Essen BioSciences) to determine cellular density within the wound relative to surrounding cellular density. When applicable, RhoA inhibitor I (C3 exoenzyme; Cytoskeleton, Inc.) was used at the indicated concentrations (see above). For siRNA-based screening, mNPCs were transfected (DharmaFECT 1) with 25 nM siRNA pools (ON-TARGETplus SMARTpool; Dharmacon RNA Technologies) corresponding to genes of interest (Supplementary Table 1). After 36 h, cells were detached and assayed for transwell migration as described above.

**Immunohistochemistry.** Immunohistochemistry for GNA13 was performed using a polymeric biotin-free horseradish peroxidase method on a Leica Microsystems Bond RXM automated stainer. The Refine polymer detection kit (Leica Biosystems) was used for detection. For antibody information, please see Supplementary Table 2.

**Immunofluorescence.** Cells were grown on polylysine (Sigma) treated and laminin coated Labtek chamber slides (Nunc), and fixed with 4% paraformaldehyde in PBS for 15 min. They were then permeabilized (with PBS, 1% BSA, 0.1% Triton X-100, and 2% serum) for 5 min and saturated with the blocking buffer (PBS, 1% BSA, and 2% serum) for 30 min. Primary (Supplementary Table 2) and secondary antibodies were successively incubated at room temperature for 1 h each. Images were acquired with a Nikon A1R MP confocal microscope. Quantifications were performed using NIH ImageJ software (http://rsb.info.nih.gov/ij/). For telomere FISH, 5 µm frozen sections were fixed with 4% paraformaldehyde for 10 min, dehydrated, and denatured at 85 °C for 5 min in hybridization mixture consisting of 10 mM Tris-HCL pH 7.2, 70% formamide, 0.5% blocking solution reagent (Roche), and a complementary Cy3-labeled peptide nucleic acid probe specific to the mammalian telomere repeat sequence TTAGGGTTAGGGTTAGGG 3′ (Applied Biosystems). Hybridization was performed at room temperature for two hours in the hybridization mixture. Slides were then washed twice with 70% formamide, 10 mM Tris–HCl pH 7.0–7.5, 0.1% BSA followed by multiple washes in PBS. Finally, slides were mounted in Prolong Gold with DAPI (Invitrogen, Grand Island, NY, USA) and examined using a Leica CTR5000 inverted fluorescent microscope.

**Protein isolation and western blotting**. Cell pellets were lysed in RIPA buffer (150 mM NaCl; 50 mM Tris pH 8.0; 1.0% Igepal CA-630; 0.5% sodium deoxycholate; 0.1% SDS; Sigma-Aldrich) supplemented with protease inhibitors (cOmplete mini, Roche Diagnostics; 1 mM PMSF; 10 mM NaF; 2.5 mM $Na_3VO_4$) and centrifuged at $10,000 \times g$ at 4 °C for 30 min. The resulting supernatant was then assessed by standard immunoblotting.

**RhoA activity assay**. Exponentially growing cells were harvested and lysed. 500 μg of protein were subjected to rhotekin-RBD bead pulldown (Rho Activation Assay Biochem Kit, Cytoskeleton Inc.) for 1 h at 4 °C. After thorough washing, the samples were boiled for 5 min in Laemmli sample buffer to detach active GTP-bound Rho and then loaded onto 12% SDS–PAGE gels for immunoblotting with an anti-RhoA antibody (Supplementary Table 2).

**RNA-Seq and data analysis**. Whole transcriptome sequencing was performed at the MSKCC Integrated Genomics Operation. Total RNA was extracted from three biological mNPC replicates at passage 5 following cre-mediated *Atrx* inactivation. Cells were cultured in proliferation conditions and total RNA isolated using the RNeasy Mini Kit (Qiagen) according to manufacturers instructions. Following ribogreen quantification and quality control using an Agilent BioAnalyzer, 500 ng of total RNA was subjected to polyA selection and Truseq library preparation (Illumina) with six cycles of PCR. Samples were barcoded and sequenced on a HiSeq 2500 in a 50 bp/50 bp paired end run using the TruSeq SBS Kit v3 (Illumina). An average of 42 million paired reads was generated per sample. At most, ribosomal reads represented 0.1% of total and mRNA accounted for 74.9% on average. Raw sequencing data was converted to gene expression values using CLC genomics workbench (Qiagen). Data were then quantile normalized (Partek), clustered, and used to derive differentially expressed gene sets (2-fold expression change, $Q < 0.01$). Enrichment and pathway analyses were performed using Panther for GO correlations, along with IPA, both with standard parameters. ssGSEA[52] was performed using publically available glioma[2,25,26] and neuroepithelial ontology[27] signatures. Briefly, gene expression values for each sample were rank-ordered and an enrichment score determined using the Empirical Cumulative Distribution Function (ECDF) of the signature genes and remaining genes. Enrichment scores were obtained by integrating the differences between signature and non-signature ECDF results. Prior to undertaking ssGSEA, principal component analysis (PCA) of transcriptional data was performed using custom R scripts. This revealed one outlier replicate from the entire sample set (Supplementary Fig. 9), which was excluded prior to further processing to optimally assess stratification between the different cellular genotypes. For neuroepithelial ontology signatures, the 500 most highly expressed genes were used. Glioma subtype-specific signatures included the 250 most highly expressed transcripts as defined by a 1-versus rest 2-class SAM[53]. Analysis of Atrx-dependent transcriptional signatures in the external expression dataset of astrocytic and oligodendrocytic precursors at different stages of development[28] was carried out as follows. For visual display as heat maps (JavaTreeView), expression profiles were centered within each time point on the average of astrocytic and oligodendrocytic groups, and biological replicates within each experimental group and time point were then averaged together.

**Quantitive PCR (qPCR)**. For RT-qPCR, RNA was prepared with RNeasy Mini Kits (Qiagen) and quality was checked by microfluidics separation using a Bioanalyser 2100. RNA was used for first-strand cDNA synthesis using random primers and SuperScript II Reverse Transcriptase (Invitrogen). qPCR was performed using Power SYBR Green PCR Master Mix (Applied Biosystems). Results were analyzed by the ΔΔCt method[54] using *Gapdh* as a housekeeping gene. Primers for RT-qPCR are listed in Supplementary Data 7, and for ChIP-PCR in Supplementary Data 8.

**TCGA gene expression data**. Normalized gene expression data were obtained from the Cancer Genome Atlas lower-grade glioma publication page (https://tcga-data.nci.nih.gov/docs/publications/lgg_2015/), and sample annotations were taken from supplementary appendix 2 of the lower-grade glioma publication[2].

**ChIP-Seq and analysis**. For ATRX ChIP[18], cells in three biological replicates cultured under proliferation conditions were fixed with 2 mM EGS (Pierce) for 45 min at room temperature in PBS. Formaldehyde was then added to 1% for 20 min and quenched with 125 mM glycine. Chromatin was sonicated with a Bioruptor Plus (Diagenode) to generate DNA fragments under 500 bp and lysates were immunoprecipitated with 40 μg ATRX H300 antibody or rabbit IgG control (Santa Cruz Biotechnology). Histone H3.3 chromatin immunoprecipitation was performed following the manufacturer instructions of Magna ChIP A Chromatin Immunoprecipitation Kit using 10 μg of Anti-Histone H3.3 antibody (Millipore). Sequencing libraries were prepared using the Kapa Hyper library Prep kit (Roche) with library amplification primer mix and sequenced as 50 base pair reads by single-read format on an Illumina HiSeq 2500 (v4 chemistry) with a maximum of 7 samples per lane (Genome Technology Center Facility, New York University). Fastq files were aligned to the mm9 genome using Bowtie2. The resulting BAM files for input and IP were subjected to peak calling using MACS2 with default parameters. For Atrx ChIP-seq, significant peaks derived from Atrx- cells were subtracted out. Promoter/gene body pie chart analysis was performed using in-house R

scripts, defining promoters as regions within 1 kb of TSSs[18]. TSSs were downloaded from the University of California at Santa Cruz (UCSC) genome browser. In all cases the most upstream TSS was used for each gene. For display, SGR files were generated from BAM files and viewed in the Integrated Genome Browser. Previously published data sets for mESCs[18] were downloaded as fastq files and subjected to identical processing pipelines as described above for mNPCs.

**ATAC-Seq and analysis**. ATAC-Seq[31] was performed on mNPCs (three biological replicates per genotype) as follows. Nuclei were prepared by two 500 g centrifugation steps, each 5 min, separated by a 5 min wash in cold PBS. Nuclei were then lysed in cold 10 mM Tris-HCL, pH 7.4, 10 nM NaCl, 3 mM $MgCl_2$, and 0.1% IGEPAL CA-630, and spun again at $500 \times g$ for 10 min. Nuclear pellets were then resuspended in transposase reaction mix (25 μl $2 \times$ TD buffer, 2.5 μl transposase (Illumina) and 22.5 μl nuclease-free water) for 30 min at 37 °C. Samples were then purified using Qiagen MinElute kits. Libraries were amplified from purified samples using customized primer sets[31] and sequenced as paired end 50 reads on a HiSeq 2500 with version 4 chemistry and 6 samples per lane (~50 million paired end reads per sample). Fastq files were aligned to mouse genome build mm9 using BWA[55]. Polymerase chain reaction duplicated reads were filtered out using the MarkDiplicates.jar Picard software tool (http://picard.sourceforge.net). Significant peaks were then identified using the Zero Inflated Negative Binomial Algorithm (ZINBA)[56]. Resulting peak intensity information was combined with peak density data using DiffBind[57] to identify differential open chromatin sites between data derived from *Atrx*- and *Atrx*+ cell lines. Novel genes with open chromatin regions were then annotated using ANNOVAR[58].

**TSS and enhancer heat maps**. TSS/promoter and enhancer regions for mouse E14.5 brain were obtained from enhancer–promoter units identified by the ENCODE project. Triplicates from ATAC-seq data were merged to produce unified aligned files and which were then transformed to heat maps using deepTools2 (version 2.2.4) with a 10 kb window[59]. For enhancers, a scale factor of 5 was used.

**Statistics**. Methods utilized for transcriptional and epigenomic analysis are described elsewhere (see above). Enrichment significance for ATAC-seq and/or Atrx ChIP-seq associated genes relative to differentially expressed transcripts was determined using the hypergeometric distribution. Analyses involving multiple datasets with replicates, including those for functional assays, were subjected to either paired or unpaired, two-tailed *t*-test.

**Data availability**. All data relevant to this study are available at request. All genomic and transcriptional data have been deposited in the Gene Expression Omnibus (GEO) repository under accession number GSE100465. Uncropped western blots appear in Supplementary Fig. 10.

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

## Acknowledgements

We acknowledge Olga Aminova for assistance with ATAC-seq sample preparation, Agnes Viale for assistance with whole transcriptome sequencing, and Brandy K. Conner for assistance with immunohistochemistry. We acknowledge Cameron Brennan for providing GSC lines. J.T.H. is supported by a Research Scholars Grant, RSG-16-179-01-DMC, from the American Cancer Society. C.R.M. is supported by NIH (R01CA204136). Support for this work also came from the Sontag Foundation (J.T.H.), the Sidney Kimmel Foundation (J.T.H.), Cycle for Survival (J.T.H.), and the Canadian Institutes of Health and Research (MOP-133586; D.J.P.). We acknowledge support form two NIH/NCI Cancer Center Support Grants (CCSGs) for MDACC (P30 CA016672) and MSKCC (P30 CA08748) and use of the Integrated Genomics Operation Core at MSKCC, with additional funding from Cycle for Survival and the Marie-Josée and Henry R. Kravis Center for Molecular Oncology.

## Author contributions

Conceptualization and design: C.D. and J.T.H.; Development and methodology: C.D. and J.T.H.; Acquisition of data: C.D., P.T.P., P.S., J.S.V., Y.W., and O.T.; Analysis and interpretation of data: C.D., P.B., P.S., M.V., Y.W., C.J.C., B.D., C.R.M., K.K., and J.T.H.; Writing, review, and/or revision of the manuscript: C.D., P.B., J.S.V., M.V., A.H., T.A.C., E.P.S, G.J.R., F.L., C.J.C., B.D., C.R.M., D.J.P., K.K., and J.T.H.; Administrative, technical,

or material support: M.V., A.H., T.A.C., E.P.S., F.L., C.J.C., B.D., C.R.M., D.J.P., K.K., J.T.H.; Study supervision: J.T.H.

## Additional information

**Competing interests:** The authors declare no competing interests.

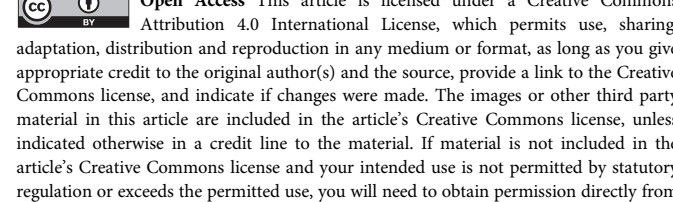

