## [Peer Review File · Nature Communications]

Reviewers' comments:

Reviewer #1 (Remarks to the Author):

In this study, the authors delete *Atrx* in murine neural progenitor cells in vitro and investigate consequent changes in cellular behavior, epigenetic changes, and gene expression patterns. They report that *Atrx* deletion results in increased motility, reduced neuronal differentiation, and enhanced astrocyte marker gene expression in *Atrx*^{-/-} *p53*^{-/-} NPCs. They also report changes in gene expression, open chromatin regions, and H3.3 binding patterns in ATRX mutant NPCs compared to *Atrx*⁺ NPCs. These results mostly validate previously published observations.

Major concerns.

Atrx is a well known modulator of the epigenome and the observed changes in the epigenome are mostly validations of previously reported interaction between ATRX and H3.3 and the role of *Atrx* in SWI/SNF complex. Furthermore, *Atrx* deletion clearly does not transform NPCs in vitro, even when combined with *p53* deletion; therefore, either *Atrx* is not a driver tumor suppressor gene or this experimental platform is not appropriate for studying its role in gliomagenesis. While it is true that enhanced invasion and GFAP expression may be directly relevant to glioma-biology, these phenotypes are not unique to glioma cells (neural cells are normally migratory and normal neural stem cells also express GFAP). On the other hand, the most significant phenotype that distinguishes normal glia from gliomas –uncontrolled proliferation– is not recapitulated by *Atrx* deletion, even when combined with *p53* deletion. In addition, several published studies already reported *Atrx* function in cell fate determination, migration, and regulation of gene expression and the epigenome (Berube et al., 2005, Seah et al., 2008, Ritchie et al., 2014, Cai et al., 2015, Drane et al., 2010, Lewis et al., 2010). Therefore, new insights gained from this study in understanding the role of *Atrx* deletion during gliomagenesis are modest.

Other concerns are listed below.

1) over interpretation of results: for example, the authors state that *Atrx* deletion “abrogated” neuronal and oligodendrocytic marker expression but figure 2B clearly shows more sustained expression of neuronal and oligodendrocytic markers in *Atrx*^{-/-} samples at d5 (although it is not clear what these expression levels are normalized to). Figure 2C,D,E also show significant expression of TUBB3 at d5 in *Atrx*^{-/-} samples. Also OLIG2 is not a unique marker of oligodendrocytes, especially in vitro. In general they should analyze more than one marker for different lineages.

2) I cannot find sufficient information about experimental design of RNA-seq analyses. Authors need to provide more details about the number of biological replicates, passage numbers, and culture conditions (proliferation vs differentiation medium) used for RNA-seq analyses. It is also unclear why they used different p-value cutoffs for different analyses– they should use a q- (FDR) value cutoff rather than p-values for RNA-seq analysis, especially if the sample size is small.

3) while the authors claim that they have analyzed *Atrx* deletion in the cell of origin for LGGs, epigenetic changes occur in cultured neurosphere in vitro. It is not clear how much of the observed epigenetic changes reflect changes in vivo due to *Atrx* deletion or more importantly, *Atrx* mutation. They should acknowledge the limitations of their experimental system in the same vein they pointed out limitations of other studies.

4) Authors claim significant differences in *Atrx* binding sites between mESC and mNPC by comparing their ChIPseq results from mNPC with those from a previously published study in mESCs. Without performing parallel studies using identical technical parameters (fixation, cross linking time, washing conditions, antibody, read depths, etc), it is difficult to interpret the significance of reported differences, especially in light of such a big difference in the total number of binding sites reported (74K in this study vs 19K in the previous study in mESC). Is this due to

experimental differences or biological differences as the authors claim? Overall, signals from mESCs appear weaker in both input and ChIP tracks shown in Figure 4e, suggesting significant differences in experimental parameters between the two studies. They should describe in detail what normalization steps or controls were used to account for these potential technical issues.

5) Numbers of genes with significant expression differences in Figures 5D and 6A do not match. This may be due to different p-value cutoffs ($p < 0.01$ vs. $p < 0.05$) used. As mentioned above, the field standard is $q < 0.05$. Authors should justify why they used different p-value thresholds for different analyses.

6) For siRNA mediated knockdown of candidate genes in Figure 6b, Gna13 is the ONLY gene showing >50% knockdown. Therefore it is not surprising that it showed the greatest effect, and singly it out as the most relevant downstream effector, when the knockdown level in other genes is not sufficient, is premature.

7) data presented in figure 6 only show correlation between Atrx status and expression of GNA13. There are multitude of differences between the different tumors that can contribute to observed differences in GNA13 expression and motility. Minimally they should test whether expression of wildtype Atrx in GS5-22 is sufficient to reduce GNA13 expression and suppress migration to establish a causal role or functional significance of Atrx-Gna13a axis in regulating migration of glioma cells.

8) authors argue that mPNCs offer "a clean epigenomic background, unlike established cancer cell lines". While genetic and epigenetic changes that occur in established cancer cell lines are previously reported, it is not at all convincing that mPNCs manipulated in vitro (in high passage cells- they report performing experiments using cells up to passage 10 in vitro) provides truly "clean epigenomic background".

9) Authors should note that neural stem cells in postnatal mouse brain express GFAP, Id3, and other markers they use as markers of astrocytic lineage. Therefore, their argument that "disease-defining molecular alteration, in and of itself, directs precursor cells of unspecified lineage toward an astrocytic phenotype at the expense of neuronal and/or oligodendrocytic differentiation" needs better support.

additional concerns:

Important experimental details are lacking throughout the manuscript such as number of biological and technical replicates analyzed for each observation, and the definition of Y-axis labels. It is often unclear how the samples are normalized and to what they are compared. Adding statistical methods used in the figure legends will be helpful.

Reviewer #2 (Remarks to the Author):

This paper analyzes effects of ATRX deficiency in murine neural precursors, describing effects on gene expression and migration, and also touching on effects of p53 mutation. Given broad interest in ATRX and its roles in cancer, the paper should be of broad, if somewhat archival, interest to the field. My comments below:

The authors show that ATRX deficiency in mNPCs led to decreased proliferation,, with additional loss of pte leading to changes in gene expression that resembled glioma subclasses, recaptulating mature astrocytic and oligodendroglial signatures, at the expense of signatures corresponding to neurons or OPCs. Curiously they go onto analyze ATRX deficiency in isolation (Figs 2 onwards) without testing how loss of p53 changes differentiation, migration, or ChIP results. Can these experiments be added?

Cross referencing genes harboring ATAC-seq open sites and Atrx enrichment peaks within 10 kb of their TSSs with transcripts associated with motility, migration, and/or invasion identified 43 genes, of which the authors focused on Gna13, a subunit of the heterotrimeric G protein G alpha 13. Ga13 levels were higher in ATRX mutant than ATRX wt gliomas, with validation in a single ATRX mutant cell line demonstrating higher levels of GNA13 and migration. This result should be validated in additional ATRX mutant lines, and by rescue with an ATRX expression construct, analyzing both expression of GNA13 and migration.

Fig 7 looks at intersections of ATRX and H3.3, in a section that is generally not well developed. Can some validation be added here?

We would like to express our sincere gratitude to both reviewers for their insightful comments. In addressing these concerns, we feel that our revised manuscript is significantly strengthened. Please find a point-by-point response to reviewer concerns below (reviewer comments italicized, responses in bold).

Reviewer #1 (Remarks to the Author):

In this study, the authors delete Atrx in murine neural progenitor cells in vitro and investigate consequent changes in cellular behavior, epigenetic changes, and gene expression patterns. They report that Atrx deletion results in increased motility, reduced neuronal differentiation, and enhanced astrocyte marker gene expression in Atrx-/- p53-/- NPCs. They also report changes in gene expression, open chromatin regions, and H3.3 binding patterns in ATRX mutant NPCs compared to Atrx+ NPCs. These results mostly validate previously published observations.

Major concerns.

Atrx is a well known modulator of the epigenome and the observed changes in the epigenome are mostly validations of previously reported interaction between ATRX and H3.3 and the role of Atrx in SWI/SNF complex.

We agree that ATRX has already been implicated in multiple aspects of epigenomic regulation. Indeed, we refer to several relevant studies in both our introduction and discussion sections. The significance of our work lies in demonstrating the extent to which ATRX-dependent epigenetic functionality influences global transcriptional profiles, and does so in a manner that reflects the pathobiology of ATRX-deficient disease states. Prior to this investigation, functional characterization of ATRX deficiency in cancer had primarily focused on the alternative lengthening of telomeres (ALT) phenotype with comparatively little attention directed toward how ATRX loss influences tumor biology through shifts in gene expression. Our work reveals that, in appropriate cellular and molecular contexts, ATRX deficiency broadly impacts transcription, resulting in characteristic, disease-relevant phenotypes. Moreover, we provide mechanistic insights into the specific epigenetic and gene regulatory alterations driving these processes. As such, we feel that our findings are both novel and highly significant. These points are included in our discussion section.

Furthermore, Atrx deletion clearly does not transform NPCs in vitro, even when combined with p53 deletion; therefore, either Atrx is not a driver tumor suppressor gene or this experimental platform is not appropriate for studying its role in gliomagenesis.

While we appreciate this comment, the fact that combined Atrx and Tp53 inactivation does not transform mNPCs in vitro is hardly unexpected and does not invalidate our experimental platform. Despite the fact that ATRX mutations occur at high rates in multiple cancers, their ability to drive tumorigenesis in experimental systems remains unestablished. To the best of our knowledge, no other group has yet been able to induce de novo tumorigenesis by inactivating ATRX, even in combination with Tp53 loss. In the case of glioma, the most relevant model published thus far layers Atrx deficiency on top of an already highly malignant NRAS-driven mouse tumor¹ (we refer to this study in our discussion section). Moreover, most ATRX-deficient human gliomas are initially low-grade, and despite their ultimately lethal behavior, grow indolently for many years, slowly acquiring additional molecular alterations over time. In light of these well-established disease characteristics, we fully expected in our studies that the oncogenic impact of ATRX deficiency, even in the appropriate cellular and molecular context, might be subtle.

We were, nonetheless, able to identify and characterize differentiation and motility phenotypes highly relevant to glioma biology while also establishing a central, gene regulatory role for ATRX in the mNPC compartment. In this way, our findings speak directly to the generalizability of the mNPC experimental system. We now address these concerns in the Discussion section of the manuscript, p. 15 as follows.

“The sufficiency of ATRX inactivation, either alone or in combination with TP53 loss, to induce proliferative gliomagenesis has yet to be demonstrated. Our work, along with that described in another recent report, supports the notion that the oncogenic effects of ATRX deficiency are more measured and may be dependent on as yet uncharacterized molecular factors. “

While it is true that enhanced invasion and GFAP expression may be directly relevant to glioma-biology, these phenotypes are not unique to glioma cells (neural cells are normally migratory and normal neural stem cells also express GFAP).

We appreciate this comment, and agree that increased cellular motility and an astrocytic lineage profile are not unique to glioma cells *per se*. Nevertheless, they are established disease-relevant characteristics and their induction through isolated Atrx loss in a multipotent neuroepithelial cellular context is itself remarkable. Please see our response to comment #9 for a more thorough discussion of astrocytic markers in glioma. The reviewer also raises interesting comparisons with normal developmental processes in the central nervous system. Indeed, the inappropriate mobilization of latent developmental pathways may underlie a significant subset of the Atrx-deficient biology we observe. We now speak to these considerations in the Discussion section of the manuscript, p. 16, as follows.

“These findings support the notion that apparent lineage derivation in cancer is a fluid concept, dependent on molecular as well as cellular context, while also suggesting that mobilization of latent developmentally relevant pathways may underlie a significant subset of ATRX-deficient biology.”

On the other hand, the most significant phenotype that distinguishes normal glia from gliomas – uncontrolled proliferation- is not recapitulated by Atrx deletion, even when combined with p53 deletion. In addition, several published studies already reported Atrx function in cell fate determination, migration, and regulation of gene expression and the epigenome (Berube et al., 2005, Seah et al., 2008, Ritchie et al., 2014, Cai et la., 2015, Drane et al., 2010, Lewis et al., 2010). Therefore, new insights gained from this study in understanding the role of Atrx deletion during gliomagenesis are modest.

We appreciate these insights from the reviewer. The inability of Atrx deficiency to induce uncontrolled proliferation is discussed extensively in response to an earlier comment (see above). With regard to functional analysis of Atrx relative to cell fate determination, migration, and gene expression regulation, we acknowledge that multiple prior studies have been conducted, as listed by the reviewer. However, we believe our manuscript represents a significant and important departure from the conclusions reached by earlier work.

Two of the listed papers focused primarily on the delineation of ATRX functionality relative to H3.3 incorporation into pericentric and telomeric heterochromatin^{2,3}. Our findings reveal that the epigenomic impact of ATRX deficiency in disease-relevant

cellular and molecular contexts is not limited to heterochromatin, but also extensively involves gene regulatory regions, with crucial phenotypic consequences.

Three other listed papers addressed neuronal phenotypes associated with ATRX deficiency *in vivo*⁴⁻⁶. However, these studies induced ATRX deficiency almost exclusively in a p53-intact molecular context, which is not seen in ATRX-mutant glioma. Moreover, the tendency of isolated ATRX deficiency to induce p53-dependent apoptosis is well documented, and the listed studies themselves reported the involvement of this mechanism in at least a significant subset of their observed ATRX-deficient phenotypes. Our work characterizes the underlying biology of ATRX-dependent disease phenotypes arising in the p53-deficient context, circumventing p53-mediated cell death to explore the more glioma-relevant consequences of ATRX deficiency. We have included additional text in the Discussion section, p 15, relating to these considerations as follows.

“Prior functional analysis of ATRX deficiency in the central nervous system has occurred primarily in the *TP53*-intact setting, which tends to result in cell death and, as such, may not represent an ideal model for glioma biology. Indeed, we also observed increased cell death in *Atrx*-intact mNPC isogenics.”

Finally, the recent paper by Cai et al. did examine ATRX deficiency specifically in glioma cells⁷. That being said, it described what was primarily a profiling study examining the differences in DNA methylation and gene expression patterns between ATRX-low and ATRX-high gliomas. The experimental work that was undertaken involved ATRX knockdown in established and extensively passaged LN-229 glioblastoma cells. It should be noted that IDH-wild type glioblastoma, the cancer variant from which LN-229 cells are derived, rarely harbors ATRX mutation. Our investigations are much more extensive from a mechanistic standpoint and, by incorporating putative cells of origin for ATRX-mutant glioma (mNPCs), we believe that their essential cellular and molecular context is more relevant to the basic biology of the disease in question. Moreover, while we note that Cai et al. found reduced and not increased motility associated with ATRX deficiency, another very recent report has confirmed our observation of increased motility in neural stem cells⁸. These considerations are now addressed in the Discussion section, p. 16, as follows.

“Prior work using LN 229 glioblastoma cells as a model system had found reduced cell migration arising with ATRX deficiency. However, a more recent report has validated our findings in neural stem cells also harboring molecular alterations (*IDH1* mutation and *TP53* inactivation) characterizing ATRX-deficient astrocytoma.”

The papers mentioned by the reviewer are also now cited at appropriate points in the manuscript.

Other concerns are listed below.

*1) over interpretation of results: for example, the authors state that *Atrx* deletion “abrogated” neuronal and oligodendrocytic marker expression but figure 2B clearly shows more sustained expression of neuronal and oligodendrocytic markers in *Atrx*^{-/-} samples at d5 (although it is not clear what these expression levels are normalized to). Figure 2C,D,E also show significant expression of *TUBB3* at d5 in *Artx*^{-/-} samples. Also *OLIG2* is not a unique marker of oligodendrocytes, especially *in vitro*. In general they should analyze more than one marker for different lineages.*

We thank the reviewer for recognizing this error in FIG. 2B. The prior legend incorrectly identified *Atrx*- traces as dotted lines and *Atrx*+ traces as solid lines (the opposite of what was intended). This typo has been corrected and now shows the proper associations between *Atrx*+ and *Atrx*- mNPCs and their respective differentiation marker expression levels over time. We apologize for this oversight. In this figure, all data is normalized to the highest expression level reached by each individual marker across the entire data set. All other measurements are scaled relative to these maximal levels. We now describe this process in the FIG. 2 legend as follows.

“In all cases, data are scaled relative to the highest expression level for each marker across the sample set.”

Regarding figures 2c-2e, we agree that *Tubb3* expression remained in *Atrx*- mNPCs at day 5. However, that level was much lower than in *Atrx*+ cells and was accompanied by reduced mature neurite length (assessed by *Tubb3* immunofluorescence). Taken together, we interpreted these findings to indicate that normal neuronal differentiation in mNPCs is impaired by *Atrx* deficiency. Finally, we did examine multiple markers of astrocytic, neuronal, and oligodendrocytic differentiation in parallel studies. These data are shown in FIG. S3a-S3c.

2) I cannot find sufficient information about experimental design of RNA-seq analyses. Authors need to provide more details about the number of biological replicates, passage numbers, and culture conditions (proliferation vs differentiation medium) used for RNA-seq analyses. It is also unclear why they used different p-value cutoffs for different analyses- they should use a q- (FDR) value cutoff rather than p-values for RNA-seq analysis, especially if the sample size is small.

We appreciate these insights and suggestions. The requested detail has been added to the Methods section, p 23, as follows.

“Total RNA was extracted from three biological mNPC replicates at passage 5 following cre-mediated *Atrx* inactivation. Cells were cultured in proliferation conditions and total RNA isolated using the RNeasy Mini Kit (Qiagen) according to manufacturers instructions.”

Additionally, we have standardized our transcriptional analysis to correspond to a Q value of < 0.01 (see response to comment #5 below)

*3) while the authors claim that they have analyzed *Atrx* deletion in the cell of origin for LGGs, epigenetic changes occur in cultured neurosphere in vitro. It is not clear how much of the observed epigenetic changes reflect changes in vivo due to *Atrx* deletion or more importantly, *Atrx* mutation. They should acknowledge the limitations of their experimental system in the same vein they pointed out limitations of other studies.*

We appreciate this comment. We chose our system on the strength of prior literature and attempted to use primary cells at minimal passaging. Moreover, we were able to validate important downstream transcriptional findings (like those involving *GNA13*) in human tissue and cell lines, providing additional support for the physiological relevance of our core experimental models. Nevertheless, we fully acknowledge that every *in vitro* system is fundamentally limited in its ability to recapitulate human disease.

Accordingly, we have modified the first paragraph of the Discussion section, p. 14, with the following sentence.

“While no *in vitro* system can perfectly recapitulate the process of oncogenesis, these primary cells, when not excessively passaged, offered an opportunity to study the gliomagenic effects of Atrx deficiency in a molecular background not extensively altered by other cancer-promoting physiological disruptions.”

We hope that this fairly acknowledges the limitations of our experimental system and are open to other suggestions regarding appropriate caveats.

4) Authors claim significant differences in Atrx binding sites between mESC and mNPC by comparing their ChIPseq results from mNPC with those from a previously published study in mESCs. Without performing parallel studies using identical technical parameters (fixation, cross linking time, washing conditions, antibody, read depths, etc), it is difficult to interpret the significance of reported differences, especially in light of such a big difference in the total number of binding sites reported (74K in this study vs 19K in the previous study in mESC). Is this due to experimental differences or biological differences as the authors claim? Overall, signals from mESCs appear weaker in both input and ChIP tracks shown in Figure 4e, suggesting significant differences in experimental parameters between the two studies. They should describe in detail what normalization steps or controls were used to account for these potential technical issues.

We appreciate the reviewer’s insights in this regard. For our Atrx ChIP-seq experiments in mNPCs, we used an antibody and protocol identical to the ones implemented in the prior mESC study⁹ and identical computational workups from the point of raw fastq files onward (see Methods). Moreover, in our validation of ChIP-seq methodology, we employed the same Q-PCR primers as were used previously for mESCs. Finally, the nearly equivalent extent of overlap (66% versus 65%; Results section, p. 9) between Atrx binding peaks and tandem repeat regions found in mNPCs and mESCs indicates that ATRX localizes to a similar core profile of physiologically relevant sites in both cellular contexts.

Nevertheless, we acknowledge that subtle differences in IP and/or sequencing efficiency may exist and their effects cannot be comprehensively ascertained without parallel studies involving both cell lines. That being said, we would expect these sorts of discrepancies to primarily impact total peak number and result largely in quantitative, rather than qualitative, distinctions between datasets. While we did find increased numbers of significant binding peaks in mNPCs relative to published results for mESCs, the qualitative differences between respective peak distributions were more striking from a functional standpoint. As described on p. 10, the mNPC binding pattern correlated with gene bodies and promoters to a much greater extent than that of mESCs, emphasizing the fundamental biological distinctions between the two cellular contexts. We now address these concerns in the Discussion section, p. 16-17 as follows.

“While the different number of Atrx binding peaks identified in our mNPC profiles may partially reflect technical variability between the two studies, the notable correlation of Atrx distribution with gene bodies and promoter regions, not seen in mESCs, speaks to a hitherto unappreciated, cell line-specific functionality for Atrx.”

5) Numbers of genes with significant expression differences in Figures 5D and 6A do not match.

This may be due to different p-value cutoffs ($p < 0.01$ vs. $p < 0.05$) used. As mentioned above, the field standard is $q < 0.05$. Authors should justify why they used different p-value thresholds for different analyses.

We appreciate this comment. The analysis described in FIG. 6 was performed separately from the rest of the paper and was initially based on a combination of $FC > 2$ and $P < 0.01$. This did not include the FDR correction ($Q < 0.01$), as had been used in earlier figures. We have corrected this oversight and now all of the transcriptional workflows described in the manuscript use the same $FC > 2$, $Q < 0.01$ cutoffs. The numbers in FIG. 6a have changed modestly as a result. We have also redone the Panther (GO) analysis presented in FIG. 2a in light of these changes.

6) For siRNA mediated knockdown of candidate genes in Figure 6b, Gna13 is the ONLY gene showing >50% knockdown. Therefore it is not surprising that it showed the greatest effect, and singly it out as the most relevant downstream effector, when the knockdown level in other genes is not sufficient, is premature.

The reviewer is correct in these observations. It was not our intention to argue that *GNA13* was the single most important mediator of the *ATRX*-deficient glioma cell migration. As the reviewer has indicated, our own data initially implicated multiple *Atrx* target genes in mNPC motility. Subsequent validation work on *GNA13* was simply intended as a mechanistic case study demonstrating that genes functionally implicated by epigenetic studies of mNPCs are also likely drivers of disease-relevant behavior in glioma. Accordingly, our conclusions in this regard were not meant to imply a singular role for *GNA13* in glioma cell migration. In fact, more recent analyses demonstrated that two other genes implicated in our focused siRNA screen in mNPCs, *AGT* and *EMILIN1*, were also similarly overexpressed in *ATRX*- glioma. These data are now presented in FIG. S6 and described in the Results section, p. 12 as follows.

“Notably, two additional motility regulators, *AGT* and *EMILIN1*, implicated in our siRNA screen were similarly overexpressed...”

and on p. 13 as follows.

“Taken together, these data firmly implicate $G\alpha 13$ and downstream RhoA signaling, along with potentially other *ATRX*-responsive genes, in the process of *ATRX*-dependent glioma cell migration.”

Additionally, these considerations are now included in the Discussion section on p. 16 as follows.

“Our analysis implicated multiple *Atrx* target genes in glioma cell motility, including *Gna13*, whose direct transcriptional upregulation and downstream signaling through RhoA GTPases promoted mNPC migration.”

*7) data presented in figure 6 only show correlation between *Atrx* status and expression of *GNA13*. There are multitude of differences between the different tumors that can contribute to observed differences in *GNA13* expression and motility. Minimally they should test whether expression of wildtype *Atrx* in GS5-22 is sufficient to reduce *GNA13* expression and suppress migration to establish a causal role or functional significance of *Atrx*-*Gna13a* axis in regulating migration of glioma cells.*

We appreciate these very reasonable concerns. To address them, we transiently restored human wild type *ATRX* (hATRX) expression in mNPCs as well as GS 5-22 cells using electroporation. In both cases, this significantly reduced cellular motility along with *GNA13* expression, despite an estimated transfection efficiency of only ~20-30%. We were also able to obtain an additional *ATRX*-mutant GSC line (GS 8-18) and, once again, found increased *GNA13* expression and migratory behavior relative to *IDH1*-mutant, *ATRX*-wild type GSCs (TS 603). Increased *GNA13* expression was also found in a third *ATRX*-mutant GSC line only capable of passage as a flank xenograft (JHH-273). Taken together these data provide further support to the conclusion that *ATRX*-deficient cellular motility in mNPCs and GSCs is mediated, at least in part, by increased *GNA13* expression. These data are now presented in FIG. 7 and FIG. S7. Moreover, we describe these findings in the Results section, p. 13, as follows.

“These relationships were recapitulated in a second *IDH1*-mutant, *ATRX*-mutant GSC line (GS 8-18) and elevated levels of *GNA13* transcript were also seen in third *IDH1*-mutant *ATRX*-mutant GSC line (JHH-273) only capable of forming subcutaneous xenografts¹⁰ (FIG. S7). Finally, we used electroporation to transiently restore human *ATRX* expression in both mNPCs and GS 5-22 cells. While our protocol resulted in only ~20-30% transfection efficiency, we were nevertheless able to demonstrate significant reversion of the migratory phenotype accompanied by a modest, but significant decrease in *Gna13/GNA13* expression in both cell lines (FIG. 7d-7j).”

Finally, we have made appropriate additions to the Methods section, p. 20, as follows.

“Transfections. Human *ATRX* was re-expressed in mNPCs and GSCs by electroporation using the plasmid IF-GFP-*ATRX* (gift from Michael Dyer: Addgene # 45444). Electroporations were performed using Nucleofector™ Kit for Mouse Neural Stem Cells (Lonza) according to manufacturer’s guidelines. 20 µg of either IF-GFP-*ATRX* (hATRX) and control pGFP (Con) plasmid were transfected in 5 X 10⁶ cells, which were harvested 48-72 hours later for RNA and protein extraction and/or migration assay.”

8) authors argue that mPNCs offer “a clean epigenomic background, unlike established cancer cell lines”. While genetic and epigenetic changes that occur in established cancer cell lines are previously reported, it is not at all convincing that mPNCs manipulated in vitro (in high passage cells- they report performing experiments using cells up to passage 10 in vitro) provides truly “clean epigenomic background”.

The reviewer is correct in these assertions and we agree that “clean epigenomic background” reflects an imprecise choice of words. We have replaced this sentence with the following on p. 14.

“...these primary cells, when not excessively passaged, also offered an opportunity to study the gliomagenic effects of *Atrx* deficiency in a molecular background not extensively altered by other cancer-promoting physiological disruptions.”

9) Authors should note that neural stem cells in postnatal mouse brain express GFAP, Id3, and other markers they use as markers of astrocytic lineage. Therefore, their argument that “disease-defining molecular alteration, in and of itself, directs precursor cells of unspecified lineage toward an astrocytic phenotype at the expense of neuronal and/or oligodendrocytic differentiation” needs better support.

The reviewer is absolutely correct that the markers used in our study are not entirely specific for astrocytic histogenesis. We also provide compelling transcriptional signature correlations with astrocytic precursors in FIG. 2f, representing an additional layer of evidence in support of our assertions. However, in a larger sense, it was not our intention to definitively prove that ATRX deficiency drives astrocyte development, but rather to demonstrate that ATRX deficiency promotes disease-relevant phenotypes, in this case expression of markers conventionally associated with astrocyte lineage, in appropriate cellular contexts.

Standard nomenclature among Neuropathologists designating *IDH1/2* and *ATRX*-mutant gliomas as “astrocytomas” derives from morphological and immunohistochemical assessment, not precise knowledge of histogenesis. In fact, it remains unclear from precisely what cells astrocytomas, or diffuse gliomas in general, actually arise (although multipotent neuroepithelial progenitor cells have been implicated in multiple studies). Accordingly, astrocytomas can be thought of simply as gliomas exhibiting morphological features of astrocytes and expressing markers, like GFAP, conventionally associated with astrocytic lineage. As such, while their precise derivation remains obscure, these tumors exhibit an “astrocytic lineage phenotype”. Our findings reveal that ATRX deficiency, the molecular alteration most tightly associated with astrocytoma, recapitulates the disease-defining astrocytic lineage phenotype in multipotent progenitors. We have now made efforts to clarify this distinction throughout the manuscript.

additional concerns:

Important experimental details are lacking throughout the manuscript such as number of biological and technical replicates analyzed for each observation, and the definition of Y-axis labels. It is often unclear how the samples are normalized and to what they are compared. Adding statistical methods used in the figure legends will be helpful.

Thank you for this comment. Additional detail has been added to the relevant Methods sections and/or Figure Legends to clarify replicate number, Y-axis identity, and normalization/scaling protocols. Statistical methods for all presented data are described in the Statistics section of the Methods with the exception of those relevant to epigenomic and transcriptional analyses. These appear in separate, designated Methods sections.

Reviewer #2 (Remarks to the Author):

This paper analyzes effects of ATRX deficiency in murine neural precursors, describing effects on gene expression and migration, and also touching on effects of p53 mutation. Given broad interest in ATRX and its roles in cancer, the paper should be of broad, if somewhat archival, interest to the field. My comments below:

The authors show that ATRX deficiency in mNPCs led to decreased proliferation, with additional loss of p53 leading to changes in gene expression that resembled glioma subclasses, recapitulating mature astrocytic and oligodendroglial signatures, at the expense of signatures corresponding to neurons or OPCs. Curiously they go on to analyze ATRX deficiency in isolation (Figs 2 onwards) without testing how loss of p53 changes differentiation, migration, or ChIP results. Can these experiments be added?

We appreciate this comment. The aim of our study was to functionally characterize ATRX deficiency in a truly disease-relevant context. While we initially explored both *Tp53*^{+/+} and *Tp53*^{-/-} backgrounds, we quickly noted that the phenotypic and transcriptional consequences of *Atrx* loss were much more pronounced in the absence of p53 (FIG. 1). As we argue in the paper, these findings likely reflected the well-reported tendency of *Atrx* deficiency to induce p53-dependent cell death in neuroepithelial precursors (FIG. S1d). Moreover, *ATR*X mutation arises in glioma almost exclusively in the p53-deficient state. Given these considerations, we felt justified in pursuing additional functional analysis primarily in *Tp53*^{-/-} mNPCs.

That being said, the reviewer is correct in his assertion that such an approach might overlook the functional impact of *Tp53* loss on disease-relevant phenotypes. As this study was focused on ATRX deficiency, we felt that an extensive analysis of *Tp53* inactivation in mNPCs was beyond its scope, and abundant prior work has already documented the phenotypic and molecular consequences of *Tp53* loss across a wide range of cellular contexts. Nevertheless, certain experiments that we did perform do shed light on what phenotypic role *Tp53* inactivation might have in our experimental systems. As would be expected from prior literature, *Tp53* loss increased mNPC proliferation and soft agar clonogenicity (FIG. S1a-S1b). However, its impact relative to *Atrx* functionality was less clear. For instance, the distribution of *Atrx* by ChIP-seq did not change with *Tp53* status (p. 9 and FIG. 4e). We also assessed the effect of *Tp53* inactivation on mNPC migration in the context of our broader investigations into *Atrx* deficiency and cellular motility. We found that *Tp53* loss induced only a mild, not statistically significant ($P=0.098$), increase in transwell migration that was dwarfed by the added effect of *Atrx* inactivation. These data are now presented in FIG. S5.

Cross referencing genes harboring ATAC-seq open sites and Atrx enrichment peaks within 10 kb of their TSSs with transcripts associated with motility, migration, and/or invasion identified 43 genes, of which the authors focused on Gna13, a subunit of the heterotrimeric G protein G alpha 13. Ga13 levels were higher in ATRX mutant than ATRX wt gliomas, with validation in a single ATRX mutant cell line demonstrating higher levels of GNA13 and migration. This result should be validated in additional ATRX mutant lines, and by rescue with an ATRX expression construct, analyzing both expression of GNA13 and migration.

As indicated above in response to a similar comment from reviewer #1, we transiently restored human wild type *ATR*X (hATR^X) expression in mNPCs as well as GS 5-22 cells using electroporation. In both cases, this significantly reduced cellular motility along with *GNA13* expression, despite an estimated transfection efficiency of only ~20-30%. We were also able to obtain an additional *ATR*X-mutant GSC line (GS 8-18) and, once again, found increased *GNA13* expression and migratory behavior relative to *IDH1*-mutant, *ATR*X-wild type GSCs (TS 603). Increased *GNA13* expression was also found in a third *ATR*X-mutant GSC line only capable of passage as a flank xenograft (JHH-273). Taken together these data provide further support to the conclusion that ATRX-deficient cellular motility in mNPCs and GSCs is mediated, at least in part, by increased *GNA13* expression. These data are now presented in FIG. 7 and FIG. S7. Moreover, we describe these findings in the Results section, p. 13, as follows.

“These relationships were recapitulated in a second *IDH1*-mutant, *ATR*X-mutant GSC line (GS 8-18) and elevated levels of *GNA13* transcript were also seen in third *IDH1*-mutant *ATR*X-mutant GSC line (JHH-273) only capable of forming subcutaneous xenografts¹⁰ (FIG. S7). Finally, we used electroporation to transiently restore human ATRX expression

in both mNPCs and GS 5-22 cells. While our protocol resulted in only ~20-30% transfection efficiency, we were nevertheless able to demonstrate significant reversion of the migratory phenotype accompanied by a modest, but significant decrease in *Gna13/GNA13* expression in both cell lines (FIG. 7d-7j).”

Finally, we have made appropriate additions to the Methods section, p. 20, as follows.

“Transfections. Human ATRX was re-expressed in mNPCs and GSCs by electroporation using the plasmid IF-GFP-ATRX (gift from Michael Dyer: Addgene # 45444). Electroporations were performed using Nucleofector™ Kit for Mouse Neural Stem Cells (Lonza) according to manufacturer’s guidelines. 20 µg of either IF-GFP-ATRX (hATRX) and control pGFP (Con) plasmid were transfected in 5 X 10⁶ cells, which were harvested 48-72 hours later for RNA and protein extraction and/or migration assay.”

Fig 7 looks at intersections of ATRX and H3.3, in a section that is generally not well developed. Can some validation be added here?

We appreciate this comment. In response, we performed additional ChIP at the *GNA13* locus in *ATRX+* (TS 603) and *ATRX-* (GS 5-22) human GSCs, finding significant differences in H3.3 composition. While this experiment was not isogenic, we believe it pairs nicely with our mNPC studies—which were isogenic—to validate these findings across species. These data now appear in FIG. 7d and are described in the Results section, p. 14, as follows.

“...focused ChIP analysis in *ATRX+* (TS 603) and *ATRX-* (GS 5-22) GSCs revealed differential H3.3 incorporation in association with *GNA13*, extending our findings into *bona fide* human disease models”

These data are also mentioned briefly in the Discussion section, p. 18, as follows.

“Findings related to *GNA13* were further validated in human GSCs, albeit in a non-isogenic context.”

Additionally, our initial studies provided further validation of our H3.3 findings as they relate to ATRX-deficient shifts in chromatin architecture and gene expression. Focused analyses of several other differentially expressed gene loci along with relevant controls are presented in FIG S8 and described in the Results section on p. 13.

References

1. Koschmann, C. *et al.* ATRX loss promotes tumor growth and impairs nonhomologous end joining DNA repair in glioma. *Sci Transl Med* **8**, 328ra28 (2016).
2. Drane, P., Ouararhni, K., Depaux, A., Shuaib, M. & Hamiche, A. The death-associated protein DAXX is a novel histone chaperone involved in the replication-independent deposition of H3.3. *Genes Dev* **24**, 1253-65 (2010).
3. Lewis, P.W., Elsaesser, S.J., Noh, K.M., Stadler, S.C. & Allis, C.D. Daxx is an H3.3-specific histone chaperone and cooperates with ATRX in replication-independent chromatin assembly at telomeres. *Proc Natl Acad Sci U S A* **107**, 14075-80 (2010).
4. Berube, N.G. *et al.* The chromatin-remodeling protein ATRX is critical for neuronal survival during corticogenesis. *J Clin Invest* **115**, 258-67 (2005).

5. Ritchie, K., Watson, L.A., Davidson, B., Jiang, Y. & Berube, N.G. ATRX is required for maintenance of the neuroprogenitor cell pool in the embryonic mouse brain. *Biol Open* **3**, 1158-63 (2014).
6. Seah, C. *et al.* Neuronal death resulting from targeted disruption of the Snf2 protein ATRX is mediated by p53. *J Neurosci* **28**, 12570-80 (2008).
7. Cai, J. *et al.* Loss of ATRX, associated with DNA methylation pattern of chromosome end, impacted biological behaviors of astrocytic tumors. *Oncotarget* **6**, 18105-15 (2015).
8. Modrek, A.S. *et al.* Low-Grade Astrocytoma Mutations in IDH1, P53, and ATRX Cooperate to Block Differentiation of Human Neural Stem Cells via Repression of SOX2. *Cell Rep* **21**, 1267-1280 (2017).
9. Law, M.J. *et al.* ATR-X syndrome protein targets tandem repeats and influences allele-specific expression in a size-dependent manner. *Cell* **143**, 367-78 (2010).
10. Borodovsky, A. *et al.* 5-azacytidine reduces methylation, promotes differentiation and induces tumor regression in a patient-derived IDH1 mutant glioma xenograft. *Oncotarget* **4**, 1737-47 (2013).

REVIEWERS' COMMENTS:

Reviewer #1 (Remarks to the Author):

This is a significantly improved revision and technical concerns are adequately addressed for the most part.

specific concern:

Why do the heatmaps in Figure 1d and 1e show only two replicates from the double mutant group while figure 1c shows triplicates? Please fix heatmaps in Figure 1d and e and show expression levels in all replicates.

Reviewer #2 (Remarks to the Author):

revised manuscript addresses issues raised in prior review

Once again, we would like to express our sincere thanks to both reviewers for their thoughtful attention to our manuscript. Please find a point-by-point response to reviewer concerns below (reviewer comments italicized, responses in bold).

Reviewer #1 (Remarks to the Author):

This is a significantly improved revision and technical concerns are adequately addressed for the most part.

Thank you.

specific concern:

Why do the heatmaps in Figure 1d and 1e show only two replicates from the double mutant group while figure 1c shows triplicates? Please fix heatmaps in Figure 1d and e and show expression levels in all replicates.

We appreciate this comment and agree that this discrepancy requires additional clarification. Unlike our other analyses of transcriptional data, which focused only on differentially expressed genes as determined by ANOVA, the ssGSEA shown in FIG. 1d-1e incorporated all genes, ranked in a unitless hierarchy from most to least expressed, to establish the significance of correlations with established signatures. Prior to performing this analysis, we assessed the degree of variation across all genes between all sample replicates by PCA. This preliminary processing step revealed an outlier within the sample set corresponding to one of the *Tp53*^{-/-} *Atrx*⁻ replicates. Since the large difference in gene space between this single replicate and the remaining samples was likely to dominate ssGSEA results, we excluded the replicate in an attempt to gain better resolution, specifically between the different *Tp53* and *Atrx* genotypes. The resulting data gave us a better sense of how changes in *Tp53* and *Atrx* status impact correlations with gene signatures for glioma and glioneuronal development. We now include our PCA analysis in Supplementary FIG. 9 and describe our analytic strategy in the Methods section, p. 25, as follows.

“ssGSEA was performed using publically available glioma and neuroepithelial ontology signatures. Briefly, gene expression values for each sample were rank-ordered and an enrichment score determined using the Empirical Cumulative Distribution Function (ECDF) of the signature genes and remaining genes. Enrichment scores were obtained by integrating the differences between signature and non-signature ECDF results. Prior to undertaking ssGSEA, principal component analysis (PCA) of transcriptional data was performed using custom R scripts. This revealed one outlier replicate from the entire sample set (Supplementary FIG. 9), which was excluded prior to further processing to optimally assess stratification between the different cellular genotypes.”

We should emphasize that exclusion of this replicate for the purposes of an isolated analysis should not be taken to mean that its corresponding expression data is “bad” or the product of a faulty run. Our three biological replicates corresponding to the *Tp53*^{-/-} *Atrx*⁻ genotype reflect the spectrum of transcriptional variance associated with ATRX deficiency, which we aimed to fully ascertain as part of the larger study. Importantly, despite this heterogeneity, our ANOVA analysis generated an extensive list of genes exhibiting significant differential expression in association with the *Tp53*^{-/-} *Atrx*⁻ genotype at a stringent Q value (0.01). Moreover, these transcriptional events strikingly correlated with key epigenomic findings and disease-relevant phenotypes in subsequent

analyses. Taken together, these results underscore the general robustness of our gene expression data.

Reviewer #2 (Remarks to the Author):

revised manuscript addresses issues raised in prior review

Thank you.